# Impact of glacier loss and vegetation succession on annual basin runoff

Evan Carnahan, Jason M. Amundson, and Eran Hood

Department of Natural Sciences, University of Alaska Southeast, Juneau, AK, USA

**Correspondence:** Evan Carnahan (elcarnahan@alaska.edu)

**Abstract.** We use a simplified glacier-landscape model to investigate the degree to which basin topography, climate regime, and vegetation succession impact centennial variations in basin runoff during glacier retreat. In all simulations, annual basin runoff initially increases as water is released from glacier storage but ultimately decreases to below preretreat levels due to increases in evapotranspiration and decreases in orographic precipitation. We characterize the long-term (>200 years) annual basin runoff curves with four metrics: the magnitude and timing of peak basin runoff, the time to preretreat basin runoff, and the magnitude of end basin runoff. We find that basin slope and climate regime have strong impacts on the magnitude and timing of peak basin runoff. Shallow sloping basins exhibit a later and larger peak basin runoff than steep basins and, similarly, continental glaciers produce later and larger peak basin runoff compared to maritime glaciers. Vegetation succession following glacier loss has little impact on the peak basin runoff but becomes increasingly important as time progresses, with more rapid and extensive vegetation leading to shorter times to preretreat basin runoff and lower levels of end basin runoff. We suggest that differences in the magnitude and timing of peak basin runoff in our simulations can largely be attributed to glacier dynamics: glaciers with long response times (i.e., those that respond slowly to climate change) are pushed farther out of equilibrium for a given climate forcing and produce larger variations in basin runoff than glaciers with short response times. Overall, our results demonstrate that glacier dynamics and vegetation succession should receive roughly equal attention when assessing the impacts of glacier mass loss on water resources.

## 1 Introduction

Glacier runoff is a dominant control on the timing and magnitude of runoff from glacierized watersheds (Hock, 2005). Short term water storage within glaciers impacts the diurnal characteristics of runoff, while intermediate term storage influences the seasonality of runoff by heavily concentrating runoff in summer months (Jansson et al., 2003) when glaciers can provide a substantial portion of streamflow even at very low levels of catchment glacierization (Stahl and Moore, 2006; Nolin et al., 2010; Huss, 2011). On annual time scales, water yields from glacierized watersheds can be more than double that of comparable nonglacierized watersheds (Hood and Scott, 2008).

Globally, more than a billion people live in watersheds that receive runoff from glaciers (Kaser et al., 2010). Within these watersheds, glacier runoff supports a wide variety of ecosystem services, including agricultural and municipal water supplies, hydroelectric power generation, stream temperature modulation, biodiversity, and fisheries (Milner et al., 2017; Cheesbrough

et al., 2009; Gaudard et al., 2016; Fellman et al., 2014; Dorava and Milner, 2000). Moreover, changes in runoff from glaciers have wide ranging implications for the structure and function of downstream aquatic ecosystems (Milner et al., 2009; Jacobsen et al., 2012). As a result, developing a quantitative understanding of how runoff from glaciers and their watersheds will be altered as glaciers continue to thin and recede is critical for predicting how the ecosystem services associated with glacier runoff will change in the future. The fact that glacier runoff is controlled by the energy balance at the glacier surface, and is thus highly vulnerable to future climate warming compared to other components of the terrestrial water budget, lends urgency to this task.

As watersheds deglaciate, annual basin runoff is hypothesized to show a transient increase followed by a decrease to a new, lower baseline value as glaciers are lost (e.g., Jansson et al., 2003; Moore et al., 2009)(Fig. 1). The magnitude of this change in basin water output can be substantial. For example, annual runoff from the Hofsjökull and southern Vatnajökull ice caps in Iceland is expected to increase by roughly 50% during the 21$^{st}$ century (Aðalgeirsdóttir et al., 2006). In contrast, late summer basin runoff in glacierized basins in British Columbia demonstrate widespread negative trends in recent decades (Stahl and Moore, 2006). The direction of the glacier runoff driven change in basin runoff is roughly a function of watershed glacier coverage with increasing basin runoff in heavily glacierized basins and decreasing basin runoff in catchments with diminished glacier coverage ($\lesssim$10%; Casassa et al., 2009). On a global scale, this trend is reflected in projections of regional glacier runoff, which show an increase in the Russian and Canadian high Arctic and a sharp decrease in lower latitude mountain basins in Asia, Europe, and South America where glacier coverage is lower (Bliss et al., 2014).

Previous efforts to evaluate the impact of glacier loss on basin runoff have focused on measuring and modeling runoff at the catchment (Moore et al., 2009; Huss et al., 2008; Huss, 2011; Stahl and Moore, 2006; Nolin et al., 2010) and regional (Stahl and Moore, 2006; Huss and Hock, 2018; Baraer et al., 2012) scale. While valuable, these case studies do not elucidate the broader geomorphological and glaciological controls that govern the hydrological responses of watersheds to ongoing glacier recession. Efforts to understand how glacier change will impact streamflow at the basin scale are also confounded by the fact that recently deglaciated landscapes are eco-hydrologically dynamic as a result of changes in evapotranspiration associated with vegetation succession. In addition to quantifying basin runoff from fixed gauging stations, a number of studies have assessed future changes in glacier runoff (e.g., Bliss et al., 2014), the amount of discharge from the receding glacier terminus. The former, which we address here, is critical from a water resources standpoint because of the static nature of hydroelectric and water collection infrastructure.

We build on previous work by using a simple glacier-landscape model to systematically evaluate how glacier recession and subsequent vegetation succession impact the timing and magnitude of variations in annual basin runoff (Fig. 1). By assuming a steadily warming climate, we focus on long-term changes in annual basin runoff and ignore seasonal and interannual variations associated with climate variability. We vary a suite of parameters within our simulations including: climate regime (maritime vs. continental), rate of climate change, basin slope, vegetation rates, and vegetation types. Our decision to focus on these parameters is guided by (i) theoretical glaciology, which indicates that glacier response to climate change depends most strongly on climate regime, rate of climate change, and basin slope (e.g., Harrison et al., 2001), and (ii) the lack of consistent relationships between climate, basin characteristics, and vegetation succession (in other words, we consider a wide range of

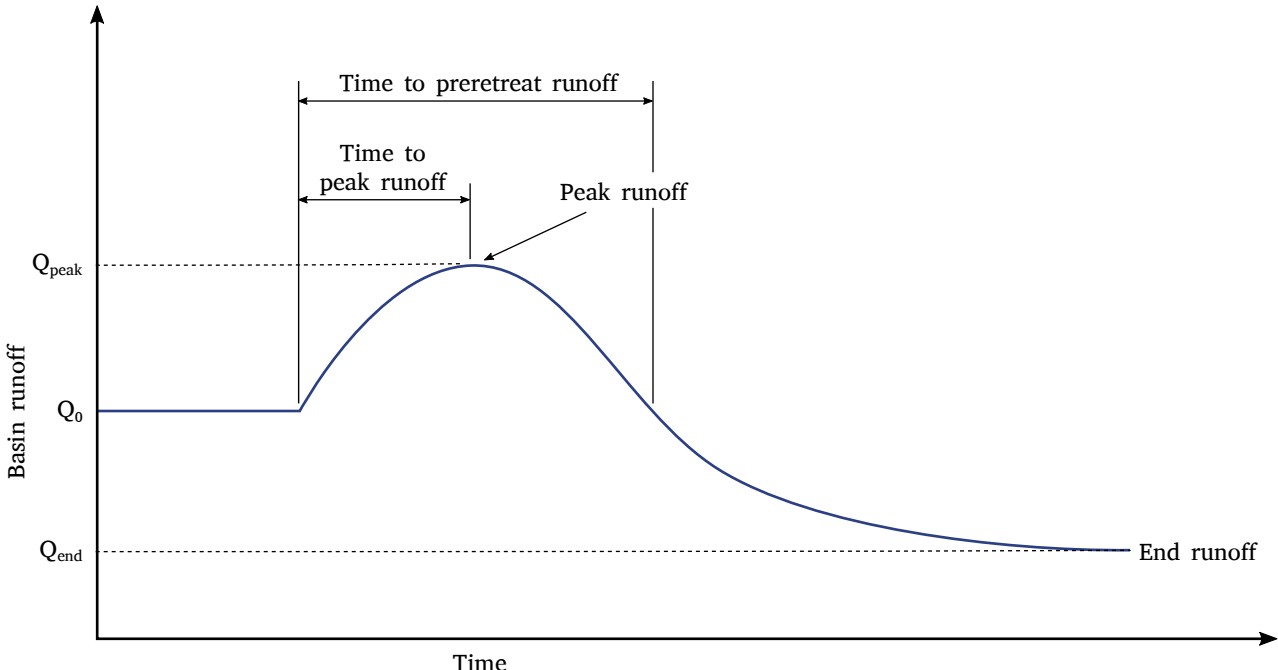

**Figure 1.** Conceptual model of Jansson et al. (2003) and Moore et al. (2009) that hypothesizes that, in response to climate warming, basin runoff will undergo a transient increase due to loss of glacier storage and a subsequent decrease past preretreat levels due to decreased glacier volume contribution and increased basin evapotranspiration.

vegetation rates and vegetation types regardless of climate regime or basin slope). We investigate the effects of each of these parameters on the long-term annual basin runoff curves, whose shapes we characterize with the following hydrological metrics: peak basin runoff, time to peak basin runoff, time to preretreat basin runoff, and end basin runoff (Fig. 1). Our findings provide insights into the hydrologic response of glacierized basins to a changing climate.

## 5  2  Methods

Assuming that nonglacier changes in water storage within a basin are negligible on annual timescales, the annual basin runoff at the watershed outlet, $Q_s$, is given by

$$Q_s = Q_g + Q_n = (P_g - Q_b) + (P_n - ET), \tag{1}$$

where $Q_g$ and $Q_n$ are the glacier runoff (i.e., the total runoff from the glacier surface; O'Neel et al., 2014) and nonglacier
10 runoff, $P_g$ and $P_n$ are the precipitation fluxes (solid plus liquid) into the glaciated and nonglaciated portions of the basin, $Q_b$ is the glacier-wide mass balance flux (accumulation minus ablation), and $ET$ is the evapotranspiration flux. For consistency, all fluxes are expressed in water equivalent units. The precipitation fluxes include solid and liquid precipitation because, over

annual time scales, solid precipitation either melts and contributes to basin runoff or is retained and contributes to the glacier mass balance. Note that the precipitation flux can be decomposed into rain and snow, and similarly the mass balance flux can be decomposed into accumulation (which equals snowfall) and melt. Consequently, the glacier runoff that we calculate using Equation (1) is identical to the sum of rain on the glacier plus glacier melt, i.e. $P_g - Q_b = P_{\text{rain}} + Q_{\text{melt}}$.

We calculate the runoff components in Equation (1) with a depth-integrated glacier flow model and a simplified landscape model. The use of a dynamic glacier model has been shown to give more accurate results for the glacier melt contribution to runoff than static models of glacier ice (Naz et al., 2014). We assume that the precipitation and mass balance rates depend on elevation and that the evapotranspiration rates are a function of time since deglaciation. The glacier flow model adjusts the elevation and length of the glacier in response to the glacier's mass balance, and the landscape model tracks the evolution of

the deglaciated landscape.

The model domain consists of a parallel-sided valley that has a constant width of 4000 m and a constant downvalley slope, with the bedrock reaching a peak elevation of 2000 m (Fig. 2a). The glacier is assumed to flow from an ice divide at the upper reaches of the valley, to span the width of the valley at all times, and to initially fill the entire length of the valley. These assumptions tend to overemphasize the relative impact of glacier runoff on basin runoff because (i) glaciers are typically wider

in their accumulation areas than in their ablation areas, which damps their response to climate change, and (ii) glaciers rarely fill an entire valley, and therefore the starting glacier runoff is generally less than the basin runoff. While simplified, the model is fast, making it possible to run numerous long-term simulations for various parameter combinations. In particular, we explore the effect that slope, vegetation rates, and vegetation types have on basin runoff over decadal time-scales under two different climate types (maritime vs. continental) and two different climate change scenarios.

**2.1   Precipitation**

Precipitation rates are needed for calculating both the glacier and nonglacier runoff ($P_g$ and $P_n$; Sections 2.2 and 2.3). We assume that precipitation varies linearly with altitude, such that

$$\dot{P}(z) = \dot{P}_0 + \frac{\mathrm{d}\dot{P}}{\mathrm{d}z} z, \tag{2}$$

where $\dot{P}$ is the width-averaged precipitation rate and $\dot{P}_0$ is the precipitation rate at sea level (Fig. 2b). For all simulations we

set $\mathrm{d}\dot{P}/\mathrm{d}z = 0.001 \text{ a}^{-1}$ (Immerzeel et al., 2015). The precipitation rate at sea level is chosen to ensure that the specific mass balance rate never exceeds the precipitation rate (see Section 2.4).

**2.2   Glacier runoff**

Glacier runoff is calculated by integrating the precipitation and specific surface mass balance rates over the glacier surface, i.e.,

$$Q_g = P_g - Q_b = \int_{\Omega_g} \left( \dot{P} - \frac{\rho_i}{\rho_w} \dot{B} \right) \mathrm{d}\Omega_g, \tag{3}$$

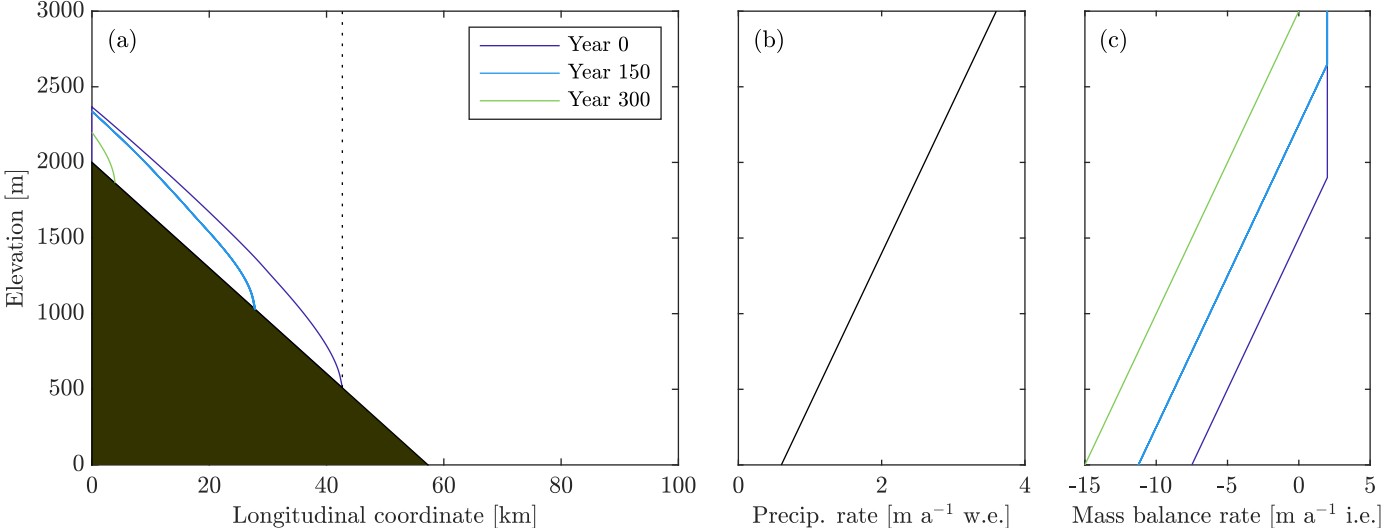

**Figure 2.** (a) Glacier thickness profiles at various stages of glacier recession with a $2°$ basin slope, continental climate, and RCP8.5 climate change scenario. The dotted vertical line demarcates the basin extent. (b) Precipitation rate with altitude, which is held constant throughout glacier retreat. (c) Specific mass balance rate with altitude at various times during glacier recession. The ELA occurs where the mass balance rate equals 0.

where $\rho_i = 917$ kg m$^{-3}$ and $\rho_w = 1000$ kg m$^{-3}$ are the densities of ice and water, $\dot{B}$ is the width-averaged specific surface mass balance rate (in units of ice equivalent) and $\Omega_g$ is the glacier surface area (in map view). The precipitation rate is given in Equation (2), and the balance rate is prescribed by using a constant mass balance gradient and imposing a maximum balance rate $\dot{B}_{\mathrm{max}}$ (as is commonly observed; e.g., Van Beusekom et al., 2010). In other words,

$$5 \quad \dot{B}(z) = \min \left( \frac{\mathrm{d}\dot{B}}{\mathrm{d}z} (z - \mathrm{ELA}), \dot{B}_{\mathrm{max}} \right), \qquad (4)$$

where ELA is the elevation of the equilibrium line altitude (ELA; Fig. 2c). We use an initial ELA of 1500 m, consistent with high- and mid-latitude glaciers (Huss and Hock, 2015). In our simulations, we vary the climate type by adjusting the balance gradient and the maximum balance rate (e.g., maritime glaciers have high balance gradients and high accumulation rates) and parameterize climate change by varying the ELA (see Section 2.4).

10     From Equation (3) it is clear that glacier runoff depends on glacier geometry (surface elevation and area), which evolves in response to mass balance and glacier dynamics. To model changes in glacier geometry, we invoke a commonly used one-dimensional, depth- and width-integrated flow model (Nick et al., 2009; Enderlin et al., 2013)(see Fig. 2a for example longitudinal cross-sections). The model is based on conservation of momentum, which requires that the glaciological driving stress is balanced by gradients in longitudinal stress, lateral drag, and basal drag (van der Veen, 2013), such that

$$15 \quad 2\frac{\partial}{\partial x} \left( H\nu \frac{\partial U}{\partial x} \right) - \frac{H}{W} \left( \frac{5U}{2AW} \right)^{1/3} - \tau_b = \rho_i g H \frac{\partial h}{\partial x}, \qquad (5)$$

where $H$ is ice thickness, $\nu$ is the depth- and width-averaged viscosity, $U$ is the depth- and width-averaged velocity, $W$ is glacier width, $A$ is the flow rate factor, $g$ is gravitational acceleration, $\tau_b$ is the basal shear stress, and $h$ is the glacier surface elevation. The viscosity depends on the strain rate:

$$\nu = A^{-1/3}\left|\frac{\partial U}{\partial x}\right|^{2/3}. \tag{6}$$

We assume a constant flow rate factor of $A = 2.4 \times 10^{-24}\,\text{Pa}^{-3}\text{s}^{-1}$, consistent with that of temperate ice (Cuffey and Paterson, 2010), and a constant basal shear stress of $10^5$ Pa, which is a typical value for valley glaciers (e.g., Brædstrup et al., 2016). In other words, we assume that the basal shear stress is at the yield stress for ice (Cuffey and Paterson, 2010). A velocity of $U = 0$ is prescribed at the ice divide ($x = 0$), and a velocity gradient is applied at the terminus by inserting the depth-averaged deviatoric stress into Glen's Flow Law. The latter is necessary because in the model the ice must maintain some finite thickness at the terminus. At each time step ($\Delta t = 0.8$ a) Equation (5) is solved for the velocity, and then the glacier surface is updated with a depth- and width-integrated mass continuity equation (van der Veen, 2013), in which

$$\frac{\partial H}{\partial t} = \dot{B} - \frac{1}{W}\frac{\partial(UHW)}{\partial x}, \tag{7}$$

and the glacier length is updated by removing any ice from the terminus that is thinner than 0.1 m.

## 2.3 Nonglacier runoff

The nonglacier runoff is calculated by assuming that the evapotranspiration rate is some fraction of the precipitation rate, such that

$$Q_n = P_n - ET = \int_{\Omega_n} C\dot{P}\,\mathrm{d}\Omega_{\mathrm{n}}, \tag{8}$$

where $0 \leq C \leq 1$ is the local annual runoff ratio (the ratio of runoff to precipitation over an area of land; hereafter referred to as simply the runoff ratio) and $\Omega_n$ is the area of the deglaciated landscape. The runoff ratio of a particular deglaciated area will vary based on the time since deglaciation. Thus, in order to calculate the nonglacier runoff, our landscape model tracks the area exposed during glacier retreat as well as changes in the surface cover as it transitions through progressively more vegetated surface types.

The landscape model is based on two simple assumptions. First, we assume that the catchment becomes increasingly vegetated following deglaciation and that the type of vegetation within a basin depends only on the time since deglaciation. The assumption is based on the time since deglaciation being highly correlated with vegetation types, biomass, and cover (Crocker and Major, 1955; Burga et al., 2010; Chapin et al., 1994; Klaar et al., 2015; Whelan and Bach, 2017; Fickert et al., 2017; Wietrzyk et al., 2018), and does not account for the effect that altitude has on vegetation levels (Cowie et al., 2014; Whelan and Bach, 2017). However, in some cases succession rates during glacier recession are comparable at different altitudes because changes in air temperature with altitude can be offset by climate warming (Fickert et al., 2017). Second, we assume that as areas of the catchment become colonized and vegetation biomass increases, the evapotranspiration rate increases until reaching

a maximum value representative of the climax vegetation state. This assumption is based on a general understanding of the processes that are expected to increase evapotranspiration, including increases in vegetation biomass, type, percentage cover, and temperature (Jaramillo et al., 2018; Andréassian, 2004; Barnett et al., 2005), although we note that there are few studies on changes in evapotranspiration throughout vegetation succession following deglaciation. Overall, results for non-glaciated paired watershed studies show increased biomass and reforestation lead to higher levels of evapotranspiration and decreased annual basin runoff (Sun et al., 2010; Klaar et al., 2015; Jaramillo et al., 2018; Bosch and Hewlett, 1982; Andréassian, 2004).

We choose to model evapotranspiration as monotonically increasing in a stepwise manner throughout vegetation succession for the following reasons. First, we are attempting to study general basin characteristics so exceptions to general rules that may cause non-monotonic increases in evapotranspiration during the transition to climax state (e.g., during the growth of eucalyptus trees; Andréassian, 2004) are of less importance. Second, the stepwise increase in evapotranspiration allows us to focus on specific stages of vegetation and not the exact transition between stages, which is less well understood compared to the change between initial vegetation and climax vegetation. We do not account for climate-driven increases in evapotranspiration because this process is attenuated in snowmelt-dominated regions (Barnett et al., 2005).

We express the change in landscape cover that occurs during vegetation succession through a stepwise parameterization of the runoff ratio. Runoff ratios range from 0.5 (forest) to ~1 (ice or rocky alpine terrain with no vegetation) and depend on the vegetation type (Andréassian, 2004; Filoso et al., 2017). We parameterize vegetation type using four runoff ratios, such that

$$
C_i = \begin{cases} C_1 & 0 \leq t \leq T_1 \\ C_2 & T_1 < t \leq T_2 \\ C_3 & T_2 < t \leq T_3 \\ C_4 & t > T_3 \end{cases},
\tag{9}
$$

where $C_i$ is the runoff ratio associated with the vegetation type $i$, $t = 0$ is the time at which a portion of the catchment is deglaciated, and $T_i$ indicates the time at which there is a transition in surface type. Thus, the runoff ratio is a function of time but varies spatially, and consequently Equation (8) can alternatively be expressed as

$$
Q_n = \sum_{i=1}^{4} C_i \int_{\Omega_{n_i}} \dot{P} \, d\Omega_{n_i},
\tag{10}
$$

where $\Omega_{n_i}$ represents the nonglacier surface area that has runoff ratio $C_i$.

## 2.4  Simulations

We use our model to test the effect that basin slope, vegetation type, and vegetation rates have on basin runoff for two different climate types and two different climate change scenarios by considering a range of parameter values. We varied the slope of the basin from shallow (2°) to steep (10°) and used six different sets of four runoff ratios, $C = \{C_1, C_2, C_3, C_4\}$, and six different sets of three vegetation timings, $T = \{T_1, T_2, T_3\}$ (see Eq. 9). The runoff ratios ranged from a high elevation or high latitude environment with no vegetation, $C = \{1, 1, 1, 1\}$, to a low elevation or low latitude environment with substantial vegetation,

$C = \{0.95, 0.8, 0.7, 0.5\}$, and the vegetation rate ranged from rapid, $T = (5\,\text{a}, 10\,\text{a}, 25\,\text{a})$, to slow, $T = (50\,\text{a}, 100\,\text{a}, 250\,\text{a})$. With the exception of the no vegetation scenario, the runoff ratio always decreased with time since deglaciation, consistent with the assumption of monotonically increasing evapotranspiration due to vegetation succession. Consequently, the runoff ratio decreases in the downvalley direction until the landscape has reached climax vegetation. Our model vegetation types and

their corresponding runoff ratios span the range of reported values for the process of vegetation succession following glacier retreat, which can be highly spatially variable even within a given climate (e.g., Crocker and Major, 1955; Burga et al., 2010; Chapin et al., 1994).

The two climate types that we define are designed to roughly mimic the climates that are experienced by maritime and continental glaciers. The climates are defined by a glacier's surface mass balance gradient and maximum surface mass balance

(Eq. 4) and by the precipitation rate at sea level (Eq. 2). For the maritime climate, we set $\mathrm{d}\dot{B}/\mathrm{d}z = 0.01\,\text{a}^{-1}$, $\dot{B}_{\max} = 4\,\text{m a}^{-1}$, and $\dot{P}_0 = 2.4\,\text{m a}^{-1}$, whereas for the continental climate these values are $\mathrm{d}\dot{B}/\mathrm{d}z = 0.005\,\text{a}^{-1}$, $\dot{B}_{\max} = 2\,\text{m a}^{-1}$, and $\dot{P}_0 = 0.55\,\text{m a}^{-1}$ (see Cuffey and Paterson, 2010, for example mass balance curves); note that the mass balance curves are defined in units of ice equivalent while the precipitation curves are defined in units of water equivalent.

In each simulation, a constant climate is used to spinup the model to a steady-state, defined as being reached when the rate

of terminus advance or retreat is less than $2\,\text{m a}^{-1}$. After reaching steady-state, the climate is changed by steadily raising the ELA (e.g., Fig. 2c). We consider two climate change scenarios that are roughly based on expected changes in ELA (Huss and Hock, 2015) under two different Representative Concentration Pathways (RCP2.6 and RCP8.5, which correspond to increases in radiative forcing of 2.6 and 8.5 W/m$^2$). In the RCP2.6 scenario we prescribe the ELA (in meters) according to

$$\text{ELA}_{\text{RCP2.6}} = 1500 + 158(1 - e^{-t/28}), \tag{11}$$

where $t$ is the time in years, resulting in an asymptotic increase in the ELA of about 150 m over a 100 year period. In contrast, during the RCP8.5 scenario we raise the ELA linearly with time:

$$\text{ELA}_{\text{RCP8.5}} = 1500 + 5t. \tag{12}$$

After the glaciers have reached steady-state we initialize changes in the ELA according to Equations 11 and 12. Neither climate warming scenario includes decadal fluctuations in climate that may complicate the simplified retreat scenario. Thus we neglect

temporal variations in precipitation and assume that changes in glacier mass balance are primarily due to warming (Fig. 2c; Van de Wal and Wild, 2001). Climate-driven changes in precipitation from snow-to-rain have equivocal effects on basin runoff, which are not included in our modeling (Neal et al., 2002; Tague and Dugger, 2010; Berghuijs et al., 2014).

During the simulations the basin is initially filled with ice (i.e., the length of the basin is defined as the initial steady-state length of the glacier, which depends on climate type and basin slope). As the glacier recedes, the portion of vegetated area

increases and previously exposed portions mature, moving through progressive vegetation types. The simulations continue until the glacier has reached a new steady-state or disappeared altogether and the newly exposed landscape has reached the final vegetation state.

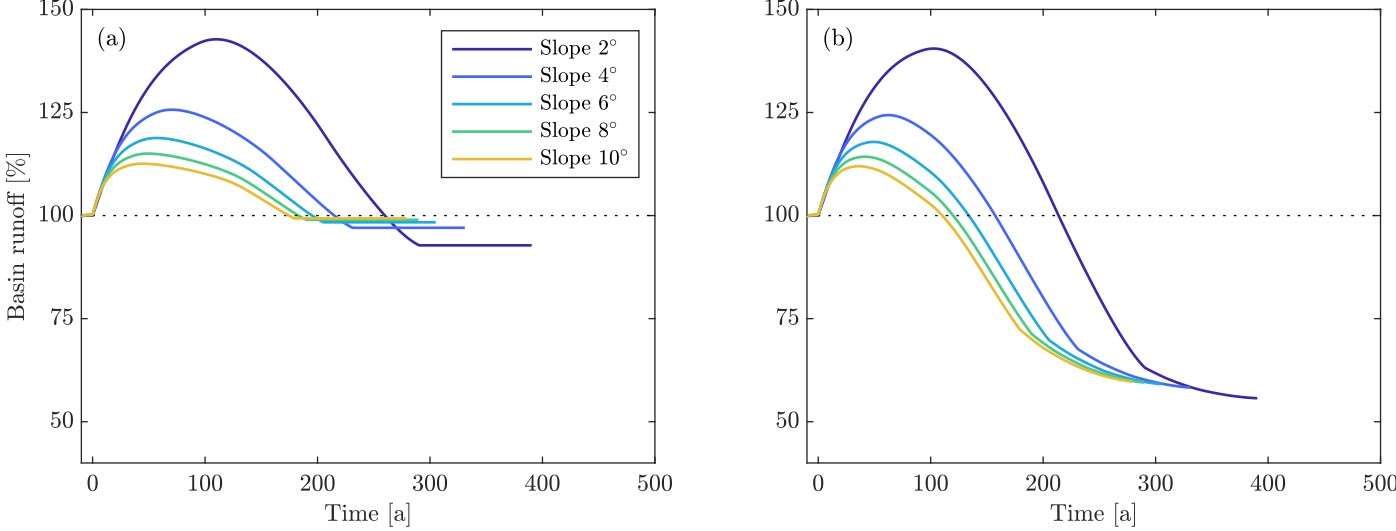

**Figure 3.** Variations in basin runoff (a) without vegetation and (b) with vegetation. For both (a) and (b) we use a maritime climate and the RCP8.5 climate change scenario. In (b) we set the vegetation timing and type as $T = \{20, 50, 100\}$ and $C = \{1, 0.9, 0.8, 0.6\}$, respectively.

## 3 Results

### 3.1 Basin runoff

Following the onset of climate change in our simulations, glaciers retreat and thin steadily until either disappearing completely (for the RCP8.5 scenarios) or reaching a new steady-state (for the RCP2.6 scenarios). Glaciers in the RCP2.6 climate scenario

lose between 16–25% of their area and 19–26% of their volume before reaching a new steady state, with steep glaciers showing higher fractional volume and area losses. Regardless of slope, fractional volume and area changes are similar (within ∼1%) between continental and maritime climates.

As glaciers retreat, basin runoff in the maritime climate under the RCP8.5 scenario experiences a transient increase of about 10–40% over a time period of 20–100 years, with shallow sloping basins experiencing substantially higher and later peak basin

runoff than steep basins (Fig. 3). Basin runoff subsequently decreases over the next 100–200 years. For simulations that include no vegetation, end basin runoff is slightly below preretreat levels due to the loss of orographic precipitation associated with the decrease in basin elevation (Fig. 3a). This result is more pronounced for shallow basins than for steep basins (-14% vs -1% for the RCP8.5 scenario) because shallow sloping basins contain longer, thicker glaciers that undergo more surface lowering. When vegetation is included, end basin runoff can fall below 50% of the preretreat basin runoff (e.g., Fig. 4). Increases in

evapotranspiration during glacier retreat partially offset increases in basin runoff driven by glacier volume loss, although this effect is small compared to the impact of vegetation on end basin runoff (Fig. 4). Rates of vegetation have no impact on the magnitude of end basin runoff because vegetation eventually reaches a climax state regardless of the rate of change (Fig.

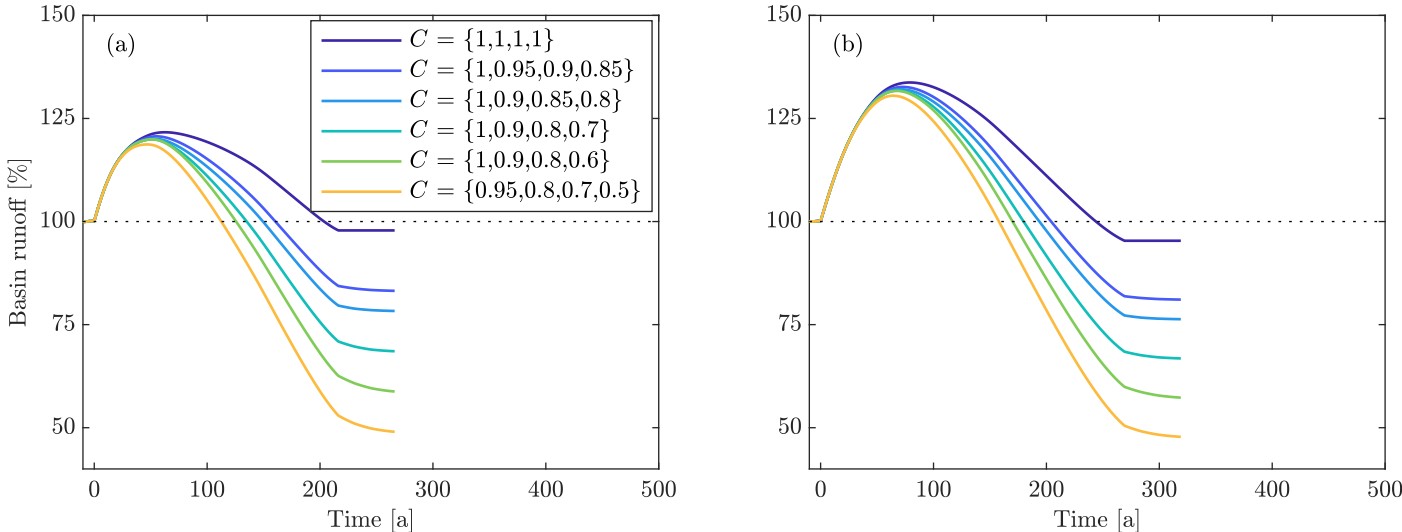

**Figure 4.** Variations in basin runoff for differing types of catchment vegetation as expressed by runoff ratios in (a) maritime and (b) continental climates. The basin slope (5°), climate change scenario (RCP8.5), and vegetation timing ($T = \{15\,\mathrm{a}, 30\,\mathrm{a}, 50\,\mathrm{a}\}$) are the same in both panels.

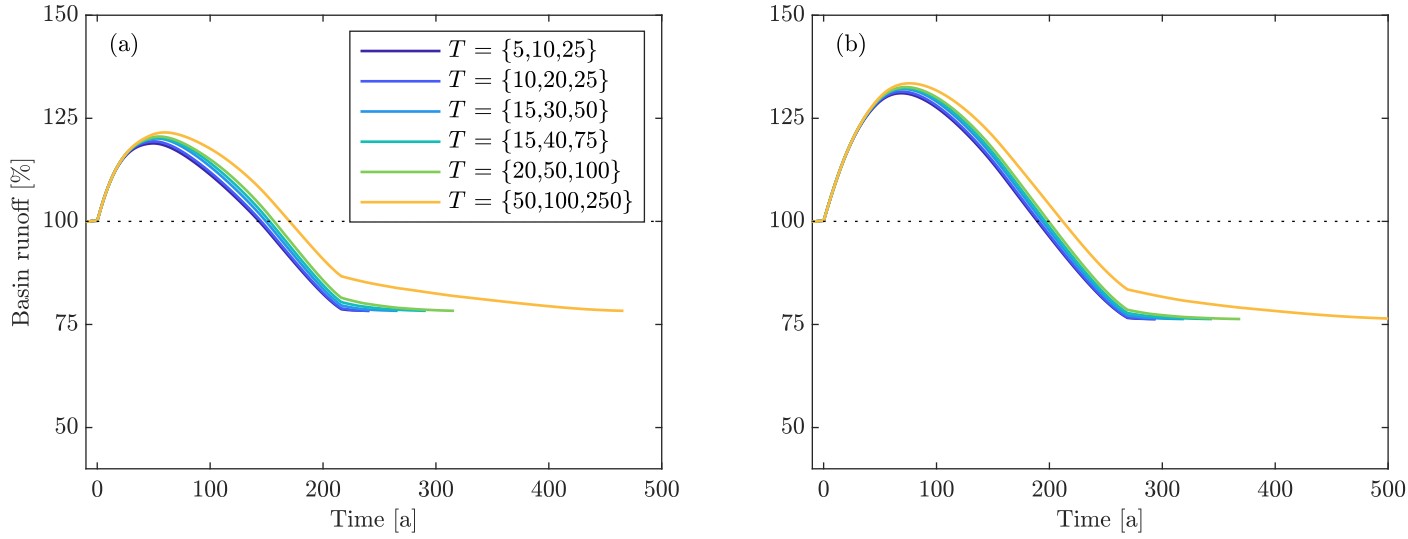

**Figure 5.** Variations in basin runoff for differing rates of catchment vegetation in (a) maritime and (b) continental climates. The basin slope (5°), climate change scenario (RCP8.5), and vegetation type ($C = \{1, 0.95, 0.85, 0.8\}$) are the same in both panels.

5). Overall, the magnitude and timing of peak basin runoff, the time to preretreat basin runoff, and the end basin runoff are minimized when vegetation occurs rapidly and progresses to a heavily vegetated state (low runoff ratio).

Temporal variations in annual basin runoff also depend on climate type. Continental basins, i.e., those with low precipitation rates and mass balance gradients, experience peak basin runoffs that are about 10–20% higher and 10–20 years later during RCP8.5 (with a greater difference for shallow basins) than comparable maritime basins, regardless of the vegetation type (Fig. 4) or rate (Fig. 5). Differences between maritime and continental basins in the RCP2.6 scenario are similar to, but smaller than, those in the RCP8.5 scenario. For example, continental basins have peak basin runoff that is $\sim$3% higher and that occurs 7–15 years later than for comparable maritime basins during the RCP2.6 scenario (not shown). Thus, changing from a maritime climate to a continental climate has a comparable effect to decreasing the slope of a basin in that it leads to a higher and later peak in basin runoff. The dependence of peak basin runoff on slope and climate type is related to the glacier response to climate change, which we discuss in Section 4.

To further quantify the effects of model parameters on basin runoff, we evaluate the relative effect that each parameter has on basin runoff by selecting a canonical set of parameters, and then vary each parameter individually around that parameter set. For the canonical set we use a maritime climate, basin slope of 5°, vegetation type of $C = \{1, 0.9, 0.8, 0.6\}$, and vegetation timing of $T = \{15\,\text{a}, 30\,\text{a}, 50\,\text{a}\}$. Within a given climate regime, the magnitude and timing of peak basin runoff is most strongly influenced by basin slope (Fig. 6a, c). In the RCP8.5 scenario, peak basin runoff from a shallow basin (2° slope) is $\sim$30% higher and occurs 70 years later than for a steep basin (10° slope). The timing and extent of catchment vegetation have minimal impact on peak basin runoff timing and magnitude, due to the limited amount of newly vegetated land that is present at peak basin runoff, regardless of the vegetation scenario. The vegetation type and rate exert an increasingly strong impact on basin runoff as time progresses. The end basin runoff is almost entirely dependent on the runoff ratio of the final vegetation state (Fig. 6b). The time to preretreat basin runoff is affected almost equally by slope and vegetation type, with vegetation rate playing a smaller but still substantial role (Fig. 6d).

Variations in model parameters consistently have a smaller impact on model output in the RCP2.6 scenario than in the RCP8.5 scenario but exhibit similar trends. This is due to the complete loss of glacier ice in the RCP8.5 scenario, which has a larger range of hydrologic impacts compared to the partial loss of glacier ice associated with the RCP2.6 scenario. One exception is the effect of varying slope on end basin runoff, which has the opposite trend for the RCP8.5 scenario than for the RCP2.6 scenario (Fig. 6b). In all simulations, glacier thinning causes a decrease in orographic precipitation that is most pronounced in shallow basins. In the RCP8.5 simulations where glaciers disappear completely, this process determines the impact of slope on end basin runoff. The situation is different for the RCP2.6 simulations because the glaciers do not disappear. There, the steep glaciers experience a larger fractional retreat than the shallow sloping glaciers, which exposes more land for vegetation and ultimately results in a slightly lower end basin runoff than for shallow basins.

Overall, peak basin runoff, time to peak basin runoff, and time to preretreat basin runoff are all smaller in the RCP2.6 scenario than in the RCP8.5 scenario. Of the four key hydrologic metrics, only the end basin runoff is higher in the RCP2.6 scenario, and this occurs because the basins do not fully deglaciate in that climate scenario. The results are similar for a continental climate, but with slightly longer times to peak and preretreat basin runoff and larger peak basin runoff (not shown).

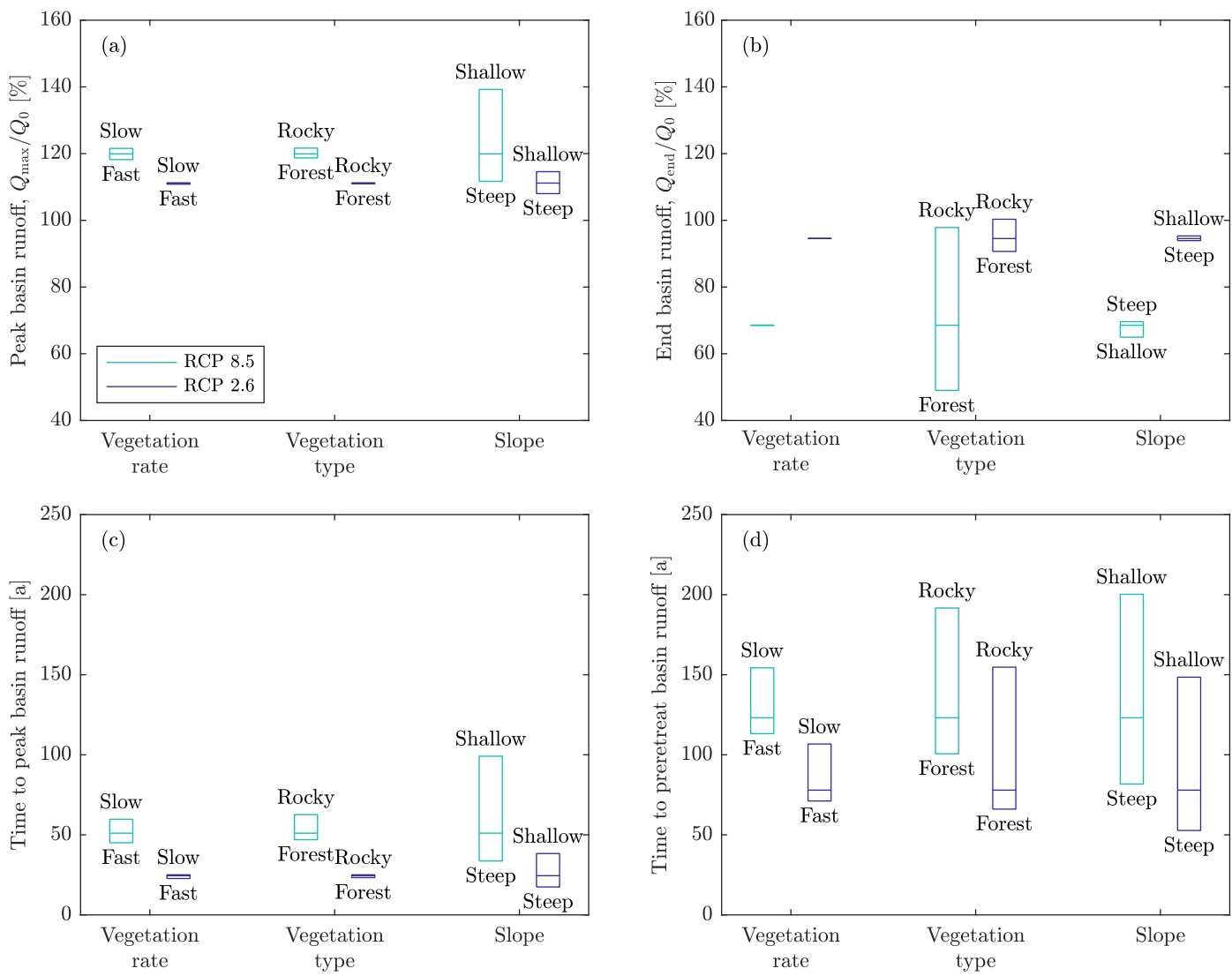

**Figure 6.** Influence of catchment vegetation and basin slope on our hydrologic metrics: (a) peak basin runoff, (b) end basin runoff, (c) time to peak basin runoff, and (d) time to preretreat basin runoff. We varied vegetation rate (slow to fast), vegetation type (rocky/unvegetated to forest), and slope (shallow to steep) across the ranges shown in the legends of Figures (3)–(5). The boxes represent the timing and relative magnitude of basin runoff associated with varying each parameter, and the lines within each box represents the value for the canonical simulation (maritime climate, basin slope of $5°$, $C = \{1, 0.9, 0.8, 0.6\}$, and $T = \{15\,\text{a}, 30\,\text{a}, 50\,\text{a}\}$).

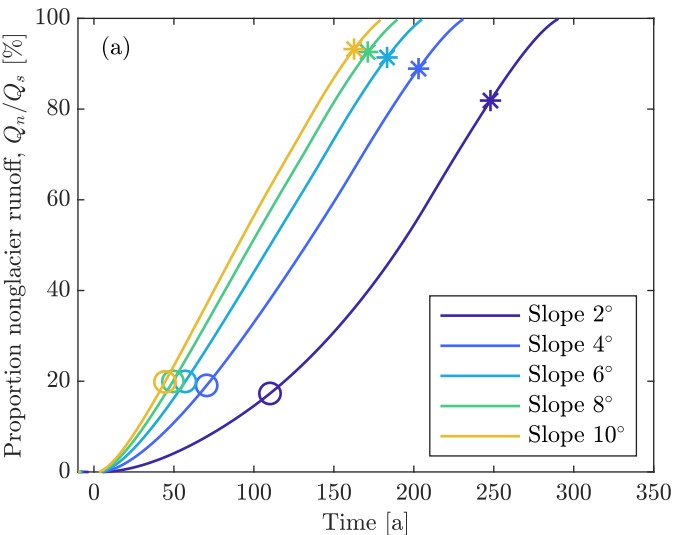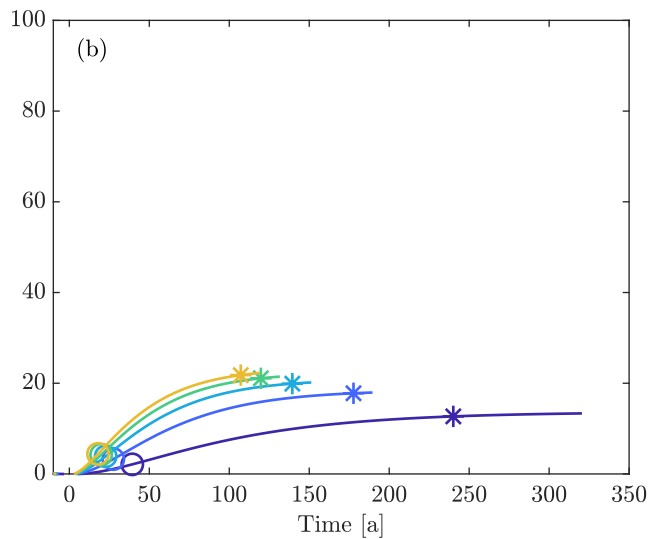

**Figure 7.** Percent contribution of nonglacier runoff to basin runoff without vegetation for a maritime climate during (a) RCP8.5 and (b) RCP2.6 climate change scenarios. Circles and asterisks indicate the time to peak basin runoff and to return to preretreat levels of basin runoff, respectively.

### 3.2 Glacier and nonglacier runoff

In our simulations, peak basin runoff and time to peak basin runoff are most strongly influenced by basin slope and climate type (Fig. 6a, c), whereas landscape evolution plays an increasingly important role in the later stages of retreat (Fig. 6b, d). The largest possible contribution of nonglacier runoff to total basin runoff occurs when vegetation and associated evapotranspiration are assumed to be negligible ($C = \{1, 1, 1, 1\}$). Under these conditions, nonglacier runoff contributes 10–20% (RCP8.5; Fig. 7a) and 1–5% (RCP2.6; Fig. 7b), of the basin runoff during peak basin runoff. By the time that basin runoff has returned to preretreat levels, the contribution from nonglacier runoff has increased to 70–95% and 9–22%, for RCP8.5 and RCP2.6 respectively (note that lower bounds are from the continental simulations and are not shown in Fig. 7). The smaller contributions of nonglacier runoff in the RCP2.6 scenario reflect the smaller amount of glacier retreat that occurred during those simulations.

Basin runoff is clearly controlled by variations in glacier runoff during the early stages of retreat. Glacier runoff, $Q_g$ (Eq. 1), undergoes a transient increase followed by a decrease to below preretreat levels (Fig. 8). This pattern is consistent for all basin slopes and climate variations. Glaciers on steep basin slopes experience lower fractional peaks in glacier runoff compared to glaciers on shallow basin slopes, with variations in slope eliciting a 35% difference in peak fractional glacier runoff for the RCP8.5 scenario (Fig. 8a) and an 8% difference in peak fractional glacier runoff for the RCP2.6 scenario (Fig. 8b). The smaller peak glacier runoff of glaciers in steep basins is also associated with an earlier peak glacier runoff. For example, in a continental climate glaciers in steep basins experience peak glacier runoff roughly 80 and 30 years before shallow sloping glaciers for the RCP8.5 and RCP2.6 scenarios, respectively. Furthermore, continental glaciers have higher peak glacier runoff, reach peak glacier runoff later, and exhibit greater variations in glacier runoff between low and high slope basins than maritime

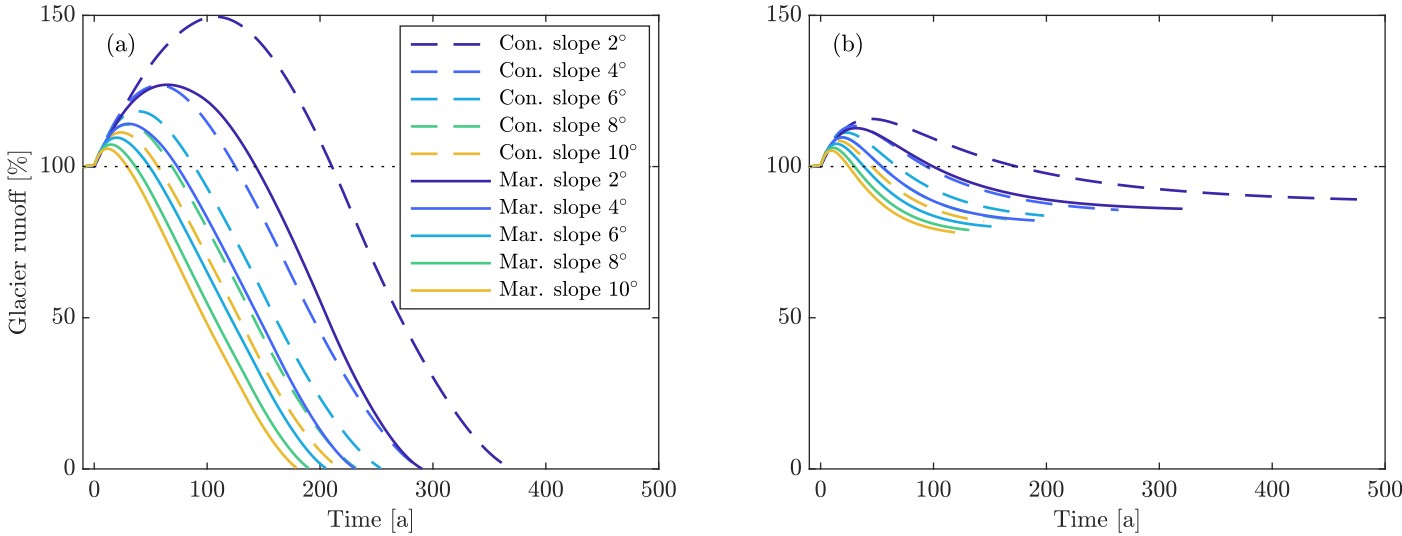

**Figure 8.** Variations in glacier runoff for differing slopes and climates during (a) RCP8.5 and (b) RCP2.6 climate change scenarios.

glaciers. The impacts of basin slope and climate type on the magnitude and timing of peak glacier runoff are similar to those observed for basin runoff, and are discussed in more detail in Section 4.2.

## 4  Discussion

### 4.1  Glacier-driven hydrological change

Our projections of the long term impacts of glacier volume loss on annual basin runoff agree closely with previous conceptual models suggesting that basin runoff will increase sharply at the onset of glacier recession, peak as glacier coverage in the basin diminishes, and then return to a steady state below the preretreat basin runoff level as the basin becomes deglacierized (Jansson et al., 2003; Moore et al., 2009). Our model results provide insights into how basin characteristics (slope, vegetation, and climatic setting) can influence the shape of this hydrologic response curve among basins with retreating glaciers. In particular,
basin slope exerts a strong control on the timing and magnitude of annual basin runoff, with steeper glaciers having a shorter time to peak basin runoff and a lower peak basin runoff than shallow sloping glaciers. Unlike peak basin runoff, final steady state basin runoff following glacier recession is strongly influenced by the type of vegetation that colonize ice-free landscapes within a basin.

    Our model results suggest that climate regime is also an important control on basin hydrological response to glacier loss,
with basins in continental climates experiencing a later and proportionally larger peak in annual basin runoff. This finding is in agreement with field measurements from a paired basin study in Alaska, which showed that glacier volume change has a strong impact on annual basin water yield in a continental environment in part because glacier volume change accounts for a larger proportion of annual streamflow in interior mountain ranges (O'Neel et al., 2014). The rate of climate warming in a

basin can similarly impact the hydrological response. In particular, applying the RCP8.5 climate scenario to our model elicited a stronger and more variable response in annual basin runoff compared to the more moderate RCP2.6 climate scenario. Much of the difference in response is due to the fact that the glacier completely disappears from the basin in the RCP8.5 scenario, resulting in a longer time to peak basin runoff, a higher peak in annual basin runoff, and a substantially lower end basin runoff

regardless of individual basin characteristics.

It is difficult to directly validate our model results due to a lack of discharge data for glacierized watersheds that span timescales comparable to those that we modeled. However, comparisons to previous studies provide insight into whether the timing and magnitude of changes in annual runoff that we modeled are consistent with runoff projections from individual glacierized basins across a range of climate conditions. At the Hofsjökull ice cap in Iceland, basin runoff is projected to peak in

about a century at 50% above current basin runoff levels (Aðalgeirsdóttir et al., 2006), which agrees well with our model results that indicate that shallow sloping maritime glaciers reach a peak basin runoff of 143% after 110 years of climate warming under RCP8.5. In Alaska, increases in summer basin runoff of 15–25% in continental and maritime glacierized catchments over a 30+ year period (O'Neel et al., 2014) are similar in magnitude to our model results for annual basin runoff. Our model results for annual basin runoff are generally lower than both Aðalgeirsdóttir et al. (2006) and O'Neel et al. (2014), possibly because

both of these studies accounted for long term increases in precipitation that we did not include in our model.

In the Pacific Northwest of North America, Nolin et al. (2010) modeled the relationship between changes in glacier extent and glacier runoff for Eliot Glacier and found that glacier runoff was reduced by 0.9% for every 1% decrease in glacier area. This ratio is highly consistent with our simulations for maritime glaciers during RCP 2.6, which showed that glacier area losses of 16–24% corresponded with decreases in glacier runoff of 14–22%. Model predictions for changes in basin runoff in Peru's

Cordillera Blanca show that basin runoff after glaciers fully disappear decreases ∼30% from present day values (Baraer et al., 2012). Given that nearly all of the basins in Baraer et al. (2012) are past peak discharge (∼30–40 years), their model results are consistent with the range of values we modelled for the decrease from peak basin runoff to end basin runoff for glaciers that fully disappear (RCP8.5) in a continental climate for all slopes and vegetation scenarios, ∼10–65%.

In a global scale analysis of future basin runoff in 56 large-scale glacier basins, the increase in basin runoff to peak basin

runoff averaged 26% for the RCP2.6 scenario and 36% for the RCP8.5 scenario (Huss and Hock, 2018), which is in broad agreement with our results. Moreover, the findings of Huss and Hock (2018) suggest that (i) there is a significant positive correlation between glacier area (shallow sloping glaciers in our study) and time to peak basin runoff and (ii) increases in the strength of the warming scenario result in a later and higher peak basin runoff. The similarity in our findings across a wide range of basin characteristics provides confidence in the trends elucidated by our model results. The delay in peak basin runoff

evident in basins with larger glaciers and basins that undergo stronger warming scenarios highlights the roles of glacier surface area and overall melt rates for determining the long term hydrological response in basins that contain glacier ice.

In many regions, glaciers have receded to the point where glacier and basin runoff have passed the peak basin runoff tipping points and are exhibiting declines in annual water output (Stahl and Moore, 2006; Bliss et al., 2014; Huss and Hock, 2018). In these basins, the variation in glacier response will no longer be the major driver of variation in basin runoff and further

examination of the ecohydrological impact of vegetation colonization is warranted. In glacierized basins with declining annual

runoff, increased evapotranspiration and canopy interception by vegetation will become an increasingly important driver of long-term variation in annual basin runoff. This finding suggests that decreases in annual basin runoff associated with glacier loss may be especially pronounced in regions such as Patagonia, New Zealand, and coastal Alaska where productive forests can rapidly recolonize newly exposed landscapes (e.g. Crocker and Major, 1955; Chapin et al., 1994).

We acknowledge that our study focused on annual basin runoff and did not explore climate-driven changes in the seasonality of basin runoff, which may be substantial even in basins where annual runoff remains largely unchanged. In particular, discharge data and model results from glacierized basins suggest that late summer basin runoff may decrease substantially with the continued loss of glacier ice (Huss and Hock, 2018; Kaser et al., 2010; Stahl and Moore, 2006; Nolin et al., 2010). Nevertheless, understanding future changes in annual basin runoff is useful for understanding the overall hydrologic response

of glacierized basins and how the wide range of ecosystem services these basins provide (e.g. Milner et al., 2017) may respond to future warming. Our findings suggest that changes in annual basin runoff with glacier loss may vary on regional scales as a result of differences in climate regime (maritime vs. continental) and regional differences in the strength of the climate warming signal. However, sub-regional variation in the hydrologic response may also be considerable as a result of catchment-scale differences in aspect, elevation, slope, and latitude, all of which influence rates of glacier ice loss and subsequent colonization

by vegetation.

## 4.2  Glacier response times

Our results indicate that the initial hydrological responses to glacier recession are dominated by variations in glacier runoff, which are themselves a result of glacier dynamic feedbacks. Thus, the peak basin runoff and time to peak basin runoff can be understood in terms of the time that it takes a glacier to respond to climate change. Theoretical work (Harrison et al., 2001)

suggests that the glacier volume response time, $\tau_v$, is given by

$$\tau_v = \frac{1}{-\dot{b}_e/H^\star - \dot{G}_e} , \tag{13}$$

where $\dot{b}_e$ is the effective specific mass balance rate in the vicinity of the terminus, $\dot{G}_e$ is the effective gradient of the specific mass balance rate with elevation, and $H^\star = \mathrm{d}V/\mathrm{d}\Omega_g$ is a thickness scale in which $V$ is volume and recalling that $\Omega_G$ is glacier surface area. The glacier response time is the e-folding time for the volume of a glacier to evolve from one steady-state to

another following a step change in climate. Equation 13 is derived from mass continuity arguments and characterizes both the timing and magnitude of the volume response of a glacier (longer response times result in larger changes in volume; Harrison et al., 2001), but depends on the assumption that $H^\star$ is constant and therefore that changes in volume are small. Nonetheless, the glacier response time is a useful tool for understanding how glacier volume and glacier runoff might be expected to evolve in a changing climate. In particular, Equation 13 indicates that glacier response times will be largest for thick glaciers (i.e.,

those that occur on shallow slopes) in a continental climate. The first term in the denominator, $-\dot{b}_e/H^\star$, will always be positive, and the larger this value the shorter the response time. Glaciers in continental climates typically have relatively small values of $-\dot{b}_e$ and thus this term tends to be small (indicating long response times). The second term in the denominator, $\dot{G}_e$, is positive and acts to increase the response time by accounting for the impact of climate on mass redistribution from high elevations to

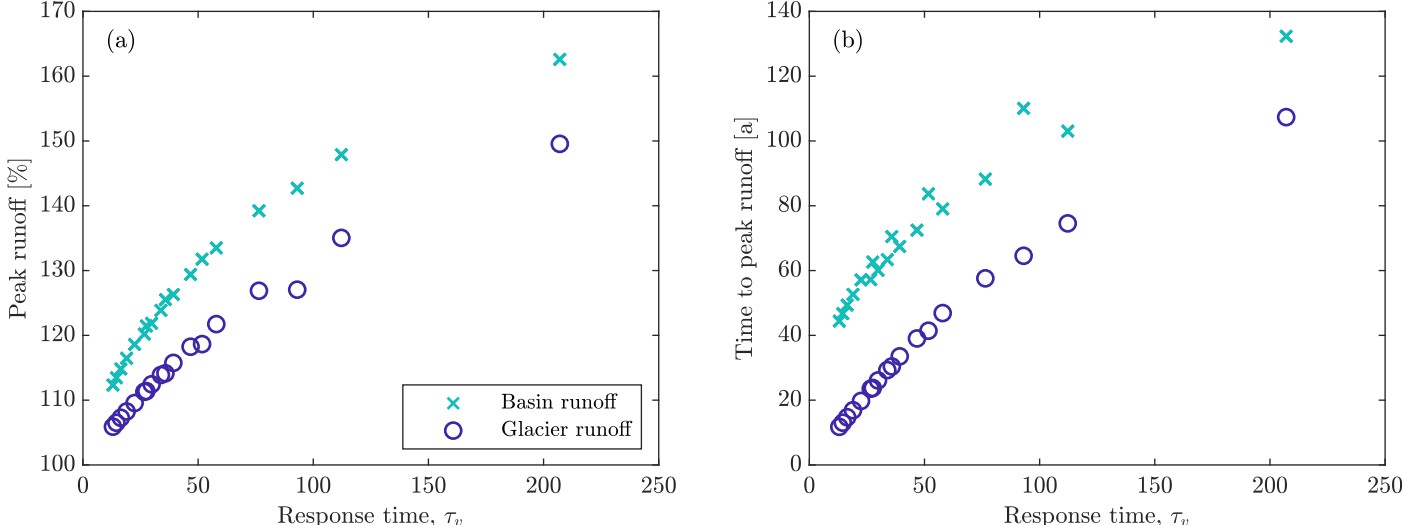

**Figure 9.** Relationships between glacier response time and (a) peak basin and glacier runoff and (b) time to peak basin and glacier runoff for all simulations that used a rocky landscape (runoff ratio of 1) and the RCP8.5 climate change scenario.

low elevations. Mass balance gradients are smaller in continental climates than in maritime climates. However, the $-\dot{b}_e/H^\star$ term is significantly smaller in continental climates, and thus the denominator is about half as small (i.e., the glacier response time is twice as long) for continental glaciers than it is for similarly sloping maritime glaciers.

Our modeling results do not follow the assumption of small changes in volume, but nonetheless they are broadly consistent
with the notion of glacier response time. We calculate the glacier response time using the initial balance rate at the terminus, the balance gradient for the respective climate, and the thickness scale $H^\star$ by calculating an average value of $\mathrm{d}V/\mathrm{d}\Omega_g$ during the first quarter of each simulation. We find that peak glacier runoff is highest and occurs latest for shallow sloping glaciers in a continental climate (i.e. those that have long response times; Fig. 8). Similarly, because variations in basin runoff are strongly influenced by glacier runoff in the early stages of retreat (Fig. 7), glacier response time is also a useful predictor of both the
magnitude and timing of peak basin runoff. We find nearly linear relationships between response time and both peak runoff and time to peak runoff for small response times (for glacier runoff and basin runoff; Fig. 9). The relationships deviate from linear because the response time calculation (i) does not directly indicate when the rate of volume loss is at a maximum, (ii) does not account for changes in basin/glacier runoff due to precipitation (it is only a statement about glacier evolution), and (iii) is based on the assumption that changes in volume are small and therefore that $H^\star$ is a constant, which breaks down as the
response time increases. A key result is that peak basin runoff and time to peak basin runoff are largest for basins containing glaciers that have long response times (Fig. 9), likely because glaciers with long response times are not able to evolve in step with climate (i.e., they have greater disequilibrium; Christian et al., 2018).

Two additional, important observations from our simulations are that (i) fractional increases in basin runoff exceed fractional increases in glacier runoff and (ii) peak basin runoff lags peak glacier runoff (Fig. 9). Although we are unable to provide precise

explanations for these differences due to nonlinear relationships between glacier volume change and associated changes in precipitation, mass balance, and vegetation rates, we can explain the trends through simple analysis of the terms affecting the basin runoff (Equation 1). First, the basin runoff and glacier runoff are both normalized by the initial basin runoff, which is equal to the initial glacier runoff, $Q_g^0$, in our simulations. The normalized basin runoff is given as

$$\frac{Q_s}{Q_g^0} = \frac{Q_g}{Q_g^0} + \frac{Q_n}{Q_g^0}. \tag{14}$$

The terms on the right hand side are the normalized glacier and nonglacier runoffs, respectively. Both the glacier and nonglacier runoffs increase during the early stages of retreat, and therefore the peak basin runoff must exceed the peak glacier runoff (in both absolute and relative terms). Second, the rate of change of basin runoff is

$$\frac{\mathrm{d}Q_s}{\mathrm{d}t} = \frac{\mathrm{d}Q_g}{\mathrm{d}t} + \frac{\mathrm{d}Q_n}{\mathrm{d}t}. \tag{15}$$

When glacier runoff reaches a peak, $\mathrm{d}Q_g/\mathrm{d}t = 0$ and consequently $\mathrm{d}Q_s/\mathrm{d}t = \mathrm{d}Q_n/\mathrm{d}t$. Peak glacier runoff occurs at a time that glacier retreat is rapidly exposing bedrock, implying that the nonglacier runoff — and by extension, basin runoff — are increasing. Thus, peak basin runoff must occur after peak glacier runoff.

## 5   Conclusions

Basin runoff varies during glacier recession due to release of water from glacier storage and subsequent colonization of
deglaciated land. Rapid glacier mass loss during the early stages of retreat drives an increase in basin runoff, which eventually decreases as a glacier shrinks and the landscape becomes increasingly vegetated. Peak basin runoff and time to peak basin runoff are largely driven by glacier response to climate change due to the major contribution of glacier runoff to basin runoff during the initial stages of retreat. Basins with glaciers that have fast response times (i.e., steep and maritime) have lower and earlier peak basin runoff because those glaciers respond rapidly to climate warming. Slow responding glaciers (i.e., shallow
sloping and continental) are unable to stay in step with climate variations and consequently experience high sustained rates of volume loss well after the initiation of climate warming, resulting in higher and later peak basin runoff. In the later stages of retreat, nonglacier runoff becomes an increasingly significant contributor to basin runoff. The time at which basin runoff falls below preretreat levels is heavily influenced by the rate of vegetation following the loss of glacier ice. Basins with fast and high levels of vegetation have earlier and lower peak basin runoff and reach preretreat levels of basin runoff substantially
earlier than those with low levels of vegetation. Basin runoff in the late stages of glacier recession is primarily determined by the extent of vegetation within the basin because evapotranspiration becomes an increasingly important term in the basin water budget compared to glacier runoff.

The model simulations that we performed were highly idealized and aimed at elucidating the fundamental controls on basin runoff over annual time scales and longer. In particular, we assumed constant glacier width, uniform basin slope, and
simplified parameterizations of climate, climate change, and vegetation succession. We also note that we began all simulations with 100% glacier cover and a glacier in a near steady-state, and consequently our simulations tend to overemphasize the

impacts of glacier recession on basin runoff. A number of processes and parameters that we either did not account for or accounted for in a simplified way have the potential to modify the shape of the annual basin runoff curves we modeled, and should be considered in future work. In particular:

- Bedrock topography, including variations in valley width and slope, will modify retreat rates, with retreat tending to be slowest when ice flows through narrow, steep constrictions.

- The formation of proglacial lakes will accelerate retreat (Larsen et al., 2007; Moyer et al., 2016) and modify basin evapotranspiration rates.

- Supraglacial debris in regions with highly erodible rock insulates glacier ice and thereby slows the rate of glacier retreat (Frans et al., 2016; Anderson and Anderson, 2016; Kienholz et al., 2017).

- Interannual variability in climate can create significant interannual variations in runoff that are superposed on the long-term basin runoff curves (O'Neel et al., 2014). In addition, interannual climate variability can push a glacier out of equilibrium with long-term climate trends (Christian et al., 2018), resulting in unexpectedly large fluctuations in runoff in subsequent years.

- In basins that have lower starting glacier cover, the effect of glacier recession on basin runoff will be dampened.

- Vegetation succession within a specific basin may differ from the simple framework we used in our model (see Section 2.3).

- We did not include groundwater storage in our model and, in some basins, the loss of glacier runoff to groundwater may be substantial (Liljedahl et al., 2017) and modify basin runoff on seasonal and annual timescales.

Increased model complexity will be required to address the full impact of climate change on the magnitude and timing of basin runoff from glacierized basins, and some variability between basins will depend on site specific factors such as bedrock topography and erodibility. The simulations that we presented here focused on what we feel are the fundamental controls on basin runoff, and as such the results provide key context for subsequent studies.

*Competing interests.* The authors have no competing interests.

*Acknowledgements.* This project was supported in part by the U.S. National Science Foundation (OPP-1504288). We thank Shad O'Neel, Brian Buma, and Christian Kienholz for discussions that led to and improved this paper.

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
