# Peer review of "Impact of glacier loss and vegetation succession on annual basin runoff"

_Hydrology and Earth System Sciences, 2018_

## Referee Comment (RC1) · B. Schaefli (Referee) · 5 Nov 2018

**General comments**

This paper proposes to analyze the joint effect of glacier retreat and revegetation (due to climate warming) on the overall water balance of glacier-covered catchments for long term evolution (up to 500 years into the future). It does so with a simplified model whose possible outcomes are studied for different glacier retreat and revegetation scenarios, for two different climate types. The studied climates are continental and maritime climates, which are emulated by adjusting the glacier mass balance rate with elevation according to observed rates in these climates. No actual data is used in the presented study but the model parameters are selected in light of known /reasonable values for existing glacier catchments.

The idea of studying the possible evolution of catchment-scale water balance resulting

from climate warming with a simplified model is appealing; it has the potential to explain in simple terms the possible outcomes (temporal increase of total basin runoff, overall decrease on the long run) without obscuring the involved mechanisms by a complex input-output model. In its current form, the results of the analysis are however hardly surprising and essentially say that "with more vegetation we get less runoff", which corresponds to an oversimplification of high alpine hydrology.

I am a hydrologist by training, with little knowledge in ice flow modelling. From my perspective, the used one-dimensional, depth- and width-integrated flow model, combined with different glacier mass balance rates seems to be a reasonable approach to generate different glacier retreat scenarios under climate warming. I find it, however, surprising that the authors choose an approach that does not allow to study the effect of the actual glacier shape (here a simple rectangle has been chosen) and that this aspect is not further discussed.

Regarding the hydrological side of the study, I have to admit that as I hydrologist I can only warn against the use of such oversimplified assumptions without sufficient discussion of the implications. To actually study the fundamental controls on the high alpine water balance, these fundamental controls and what we know thereof should be reviewed in detail before building a model.

My critic is the following: The parameterization of the effect of colonization is summarized by **two simple assumptions**: "First, we assume that the catchment becomes increasingly vegetated following deglaciation and that the type of vegetation only depends on time since deglaciation. Second, as areas of the catchment become colonized, the rate at which water is evapotranspired increases until reaching a maximum value representative of the climax vegetation state." **While the first assumption seems reasonable** (some references would certainly be useful), the second assumption omits an important body of hydrological literature of the effect of vegetation on the water balance, and in particular the effect of forest (e.g. Andreassian, 2004). Forests show typically increased ET fluxes during younger states

as compared to the climax state. Whether the typical vegetation succession to be expected in glacier catchments leads to a continuous ET increase with vegetation cover increase, remains to be demonstrated. I am not aware of literature on this topic (but it might well exist of course). In general the evolution of hydrological / geomorphological / pedological processes in moraines (and related runoff processes) can be assumed to be still largely unknown (see an ongoing project description here: http://gepris.dfg.de/gepris/projekt/318089487?language=en).

I do think that the approach is interesting. The hydrological assumptions should however be a bit more elaborate, including good references for glacier catchments and a detailed review of what we know today about the evolution of the water balance of newly vegetated areas in such catchments. If no sufficient literature can be found, possible hypotheses should be discussed in detail. This literature review should also include the important ongoing discussion what the effect of decreases in snow to rainfall ratios has on the catchment-scale water balance (Berghuijs et al., 20014). The relative decrease of snowfall might significantly contribute to the reduce of basin-scale runoff (add to the effect of vegetation). Similarly, a topic that should be discussed (even if not included in the analysis) is the interaction between glacier retreat and groundwater recharge. Not much is known so far about this topic but glacier retreat might change the relative amount of water that is available to vegetation in the non-glaciated part.

To summarize, to increase the value of this study, I suggest a good literature review of the impact of glacier retreat and the associated reduction of snow- to rainfall ratio on the water balance of high alpine catchments. Based on this, key processes and their synergy and possible unknowns should be identified. Based on this, the hydrological model can either be kept as is (but with more realistic future scenarios) or be refinde. At the very least, the hydrological simplifications should be more explicitly discussed.

**Detail comments:**

- Regarding the future ET fluxes, the reference to a paper that studied forest versus

crop / pasture across the globe in non-mountain environments (Zhang et al., 2001) is probably not adequate.

- The concept of **"runoff ratio"** is an engineering concept that was developed to separate precipitation into surface runoff and infiltration at the event scale (e.g. for the application of the so-called rational formula). What is used in this model is the **"annual runoff ratio"**, which is the ratio between total basin runoff and the total incoming precipitation. The total basin runoff is the sum of direct surface runoff and fast and slow subsurface runoff processes (and not the "runoff over an area of land"; the latter are the result of soil – vegetation interactions and groundwater recharge / release processes. This should be clear to avoid confusion for non-hydrologists.

- the conclusion should give clear indications about what should be explored on the hydrological side (not just the glaciological side)

**References**

Andreassian, V.: Waters and forests: from historical controversy to scientific debate, Journal of Hydrology, 291, 1-27, 10.1016/j.jhydrol.2003.12.015, 2004. Berghuijs, W. R., Woods, R. A., and Hrachowitz, M.: A precipitation shift from snow towards rain leads to a decrease in streamflow Nature Climate Change, 4, 583–586, 2014.

———————————————

---

## Referee Comment (RC2) · Anonymous Referee #2 · 5 Nov 2018

**Summary**

In this study the basin (glacier) peak water trajectory, following glacier retreat, is modelled using a glacier flow model in combination with some parameterizations, to simulate glacier retreat and changing vegetation in the non-glacierized areas of the basin. The effects of basin slope, climate type (maritime and continental), vegetation rate and type, and climate change scenario (RCP2.6  RCP8.5) on this trajectory are tested. The results show that slope and climate type influence the magnitude and timing of peak water, and this is related to the glacier response time. A continental climate and shallow slopes cause a higher increase in basin runoff and a later time of peak runoff, compared to a maritime climate and steep basin slopes. The effect is more pronounced in the RCP8.5 scenario. Vegetation rate and type is influencing how fast runoff levels decrease after peak water to pre- peak water runoff levels and

vegetation type determines how much runoff drops after peak runoff compared to initial runoff levels.

The modelling approach is rather mathematical, in contrast to many other glacio-hydrological studies published in HESS. This allows to perform an interesting sensitivity study, which is of interest for the HESS community. However, the more glaciological way of describing a glacierized hydrological system as presented in this study, requires more clarity, explanation and discussion when publishing in a hydrology journal (HESS). Please find my explanation, together with some other concerns below. Apart from that, the manuscript is generally well written and the figures are nicely presented.

**Major issues**

*1. Modelling framework*

The study uses a simple glacier flow model in combination with parameterizations of runoff ratios to model vegetation succession in the non-glacierized parts of the basins. Together with some climate "input", this is coupled to calculate basin runoff, glacierized runoff and nonglacier runoff over time. However, the problem is that the description of the model in the different equations and sections is not well connected (e.g. how the modelling of glacier dynamics is connected to the calculation of Qg or in which equations parameters are changing (apart from C and T)). This is important to better understand and interpret the results.

Equation 1 gives a good overview of the main modelling framework. However, from the other equations given in the methods section it is not always clear how they fit the calculation of the total basin runoff. The description of the precipitation input is sometimes a bit confusing. Why is it a separate section? And why is there written that it includes the solid and liquid fluxes? This is a bit confusing since there is no temperature input involved. Maybe it should be also made clear that precipitation "input" is constant every year. Precipitation is in this study also not a real input to

e.g. the glacier, because the mass balance is another parameter partly independent of precipitation. Please clarify the sentence "precipitation at sea level is chosen to ensure that the precipitation at elevation exceeds glacier accumulation rates". Does it mean that precipitation should fit the mass balance rates above zero? And what is the exceeding precipitation assumed to be? Can this be indicated in Figure 1?

In the section about the glacier runoff and the glacier model, it might be more clear when the section starts with the description of the glacier model and then show that the output of this glacier model (surface area of the glacier $\Omega$ g) is used to calculate the glacier part of the total basin runoff (and how it influences $\Omega$ n in the next section). Why is P(z) written in equation 2, but is "(z)" left out in equation 4? What does "min" indicate in equation 4? And why is there a maximum mass balance (Bmax)? Why can P increase with height but B not? What is meant with glacier hypsometry (L21 P4)? If it refers to equation 3 it only indicates length changes (since the width is constant), or does it also include the glacier thickness due to "z" in P and B? What does small h mean in equation 5? Is this the slope? What is solved from equation 5? And how does it relate to equation 7? I think some more explanation here would be beneficial.

The "t=0" in L1 P6 is a bit confusing with the later explanations that the climate in the model is kept constant during the first 10 years. What is t(0) in this case? Start of the simulations or when a portion of the catchment is deglaciated? Related to that it is also confusing that it is written that the climate is kept constant (for each climate type?) to reach a steady state (spin-up) and is then changed by changing the ELA, but then the climate is held constant for 10 years (no change in ELA)? Please reorder. Also the definition of "constant climate" (L25 P6) only becomes clear later in the text when it is explained that climate change is modelled by changing the ELA. The last sentence of the methods also requires some more explanation, that the simulations continue until the glacier have reached a new steady-state. How can the glacier reach a steady state when the ELA is increasing every timestep, especially in the RCP8.5

scenario? Is there a maximum ELA?

*2. Clarity*
Apart from the methods model description also other parts of the manuscript sometimes lack clarity:

- It would help if the key metrics described in the results are indicated in a conceptual figure. Especially the time to pre-retreat basin runoff and end basin runoff would get more clear from such a graph

- The "Thus" sentences in the manuscript are not always straightforward:

  – "thus the basins do not have the same length" (L8 P7) – this depends on climate type (and thus mass balance gradient) but also on slope? It would help if the initial glacier areas/lengths and volumes for all simulations (climate type and slopes) are given, together with their change over time. In that case the fractional volume changes e.g. for steep glaciers and the different climate types could be better interpreted. Why is for example the fractional volume and area change similar for both climate types, but the peak runoff differently – due to a larger volume in the continental climate? It also helps to visualize that there is a limited amount of newly vegetated land at peak runoff. It would be good to indicate/explain differences in glacier geometry due to climate type and slope in the results or methods based on the equations, e.g. why shallow sloped basins contain longer glaciers.

  – "Thus the model results tend to overemphasize the relative importance of glacier runoff on basin runoff" – because in reality one does not start with 100

  – "Thus we assume that the basal shear stress is at the yield stress" – please explain the "thus"

[Figure]

- In the results section: why are results sometimes explained for one of the two climate scenarios only?

- What is meant with glacier geometries? Slope, length, thickness?

- What is the reason that glacier runoff peaks before basin runoff? The decreases in precipitation on glaciated land also influence basin runoff?

- What is magnitude in case of end basin runoff? The magnitude is smallest for peak basin runoff, but largest for end basin runoff in case of a heavily forested state? (P8).

*3. Structure*
The introduction section of this manuscript lacks the description of a clear knowledge gap. It should be emphasized more what is new about this study (landscape coupling?) and what we do not yet know. The results section includes quite some interpretation, and even refers to the discussion (glacier response times). The results section also includes text about key metrics that should shift to methods.

*4. Discussion and implication*
In the discussion the hydrological changes (changes in annual runoff) are discussed together with their controls and compared to other literature. However, the implication of the quantitative analysis (as presented in the introduction) is lacking. What do the numbers mean and how can they be transferred to glacierized catchments around the world? Some numbers are compared, but it is not always clear which part of the graphs (trajectory) agree with observations. The simulations all start with 100% glacier cover, but what can we learn from that when a catchment has e.g. 50% glacier cover? Will it have the same variations? And what if the glacier hypsometry has not a fixed width? Why has a 1D model been chosen? Has t(0) been in the past for glacierized catchments and can we expect a similar peak runoff and rate of decline in annual

runoff? Is e.g. the size of the glacier modelled in this study representative? Other aspects that could be more emphasized is the drop of annual runoff below pre-retreat levels, which is e.g. not found/modelled in other studies (e.g. Huss Hock, 2018). Also the importance of including vegetation could be more stressed and compared with other studies (where it is often neglected).

Also the glacier response time is discussed, as an explanation why slope and climate type influence the hydrological response. Why is peak basin runoff related to the time a glacier needs to responds to climate change? This would only be half way (the time it needs to reach a new steady state)? Can the different simulations for which a response time is calculated also be indicated in Figure 8? The conclusions that are drawn in the text can now not be seen in the Figure. Is the response time – peak runoff relation also influenced because the ELA increases every time step?

**Specific remarks**

L7 P1: "rate of climate change" – what does rate mean here? Scenario might be more clear
P1 abstract: "Peak basin runoff" – use magnitude of peak basin runoff as in rest of paper to be more clear
L24 P1: "Moreover, changes in runoff...ecological function of downstream aquatic ecosystems" – The order of the sentences is strange here, because one first reads that changes in glacier runoff only effect the downstream aquatic ecosystems, but on the next page it is described how all the ecosystem services will be affected by changing glacier runoff.
L2 P2: "Glacier runoff....water budget" – this sentence does not fit here, move up or connect better
L5 P2: "lower baseline" – only Moore et al. (2009) show a lower baseline, Jansson et al. (2003) not. Also Huss Hock (2018), for example, show no lower baseline. So either

explain why there is a lower baseline, or leave it out in the introduction and discuss the differences presented in the literature in the discussion (or discuss in the introduction)

L7 P2: "increase roughly 50 percent by end of century" – compared to what?

L11 P2: "On a global scale….South America" – be more explicit here, Arctic, Canada and Russia have higher glacier coverage basins? In Asia, Europe and South America glaciers have retreated and therefore lower glacier coverage?

L14 P2: How can "Stahl and Moore (2006)" be both cited as a study on individual catchments and on regions? Nolin et al. (2010) is a study on a specific catchment so why mentioned as a study focused on the regional scale? Huss and Hock (2018) is a global scale study? "case studies" in the next sentence does not fit all of the references mentioned here.

L18 P2: what does "also" mean here?, same for "also" in line 21?

L21 P2: reduce the use of "the fact that" throughout the manuscript

L25 P2: "annual basin runoff" is used mostly in the paper, but in title and introduction "water yield" is used – why?

L1 P3: "definition 5" – please explain

L4 P4: notation of variables with an overdot to indicate width average – is overdot usually not used to indicate a derivative?

L7 P3: "precipitation at elevation" – which elevation?

L8 P5: "timestep" – indicate that timestep is one year

L15 P5: "runoff ratio (the ratio of precipitation to runoff over an area of land)" – switch precipitation and runoff -> the ratio of runoff to precipitation

L24 P5: "runoff ratios range from 0.5 (forest) to close to 1 (ice)" – on the next page it is written that runoff ratios are 1, and that it represents rocky high elevation environment with no vegetation?

Eq. 11 and 12 P6 and P7: indicate (e.g. as subscript) that equation is for RCP2.6 and the other for RCP8.5

L8 P7: "As the glacier recedes", add comma

L3 P9: "on slope and climate type and is related to the glacier response" – remove

"and" or is another variable forgotten here?

L10 P9: Fig 5a,b – this should be figure 5 a and c – see also other references to Figure 5 in this part of the results

L9 P11: "slightly longer times" – longer times of what?

L5 P12: "for all glacier geometries"- what is meant here? Slope?

L5 P13: "final steady state basin runoff following glacial recession is strongly influenced by the rate and type of vegetation" – do you mean here the final steady state basin runoff or also the timing of the end basin runoff? In the first case, this sentence contradicts the results

L15 P13: "longer response time" – what is response time here?

L29 P13: "end glacier runoff" – what is end glacier runoff?

*Figures:*

- Fig. 1:

  – Can you indicate ELA in fig. 1c?

  – For clarity it might help to also plot the lines for a maritime climate and if possible also for the RCP2.6 scenario

- It would be helpful to have the same x and y axes in all figures, since for the interpretation of some results one needs to look at several graphs

- Why is legend in some figures in the right graph and in others in the left graph?

- Please indicate the degree symbol in the "slope" legends

- When looking at the figures it is not directly clear what is compared in the left and right graphs, although it is indicated in the figure captions. Could the figures get a title or a label in the graph so it is clear what is compared in both?

- Fig. 2 :

    – Why is y axis starting at 0, but at 70 in figure 4?

    – What determines the length of the (horizontal) line indicating after peak runoff in figure a? I assume glaciers have disappeared and since no vegetation is present in figure 1 no final vegetation state needs to be reached

- Fig. 3 and 4 and 5: why is the basin slope 5 and does figure 2 not show a slope of 5 degrees?

- Fig. 5: Missing in caption, results are only shown for maritime climate?

- Fig. 6:

    – add symbols as legend

    – What determines the end of the simulation in both graphs? Compared to Figure 2a the results stop earlier in Fig. 6a. Also for 6b this is not clear

---

## Referee Comment (RC3) · Anonymous Referee #3 · 6 Nov 2018

This manuscript addresses the "peak water" concept associated with glacier response to climatic warming. As reviewed in the introduction to the manuscript, this concept was described in two review articles and has been studied empirically in a number of site-specific studies. Although the empirical studies generally confirmed the conceptual model in broad terms, two fundamental questions arise from this body of literature: (1) what is the time scale over which the "peak water" cycle progresses, and (2) does the trajectory ultimately lead to reduced runoff.

To address these questions, the authors combined a numerical model of glacier dynamics with a parameterized model of vegetation succession and its influence on runoff. They applied the model to glaciers within simplified valley geometries for scenarios representing various combinations of bed slope, vegetation type, and rates of vegetation development for two different climate types and two different climate change scenarios.

[Figure]

The simulations confirmed that basin runoff ultimately decreases relative to pre-warming conditions. For scenarios without vegetation development, this decrease results from the surface lowering associated with glacier thinning and retreat, and the subsequent reduction in precipitation. Development of vegetation results in greater reductions in basin runoff. The magnitude of and time to "peak water" were greatest for continental glaciers with shallow bed slopes and lowest for steep maritime glaciers.

Overall, this is an interesting and relevant study. However, the conclusions, at least in qualitative terms, could have been deduced fairly directly from the underlying assumptions and basic knowledge of glacier dynamics. I believe that some further analysis and more detailed consideration of vegetation dynamics and ecohydrology would strengthen the contribution of this work. Some specific comments follow.

1. There are additional processes by which annual runoff would decline in a warming climate that are not accounted for in the model. First, as pointed out by another reviewer, recent literature suggests that a shift from snow to rain results in decreased runoff even with no change in the amount of precipitation. Second, increasing air temperatures would be expected to increase evapotranspiration, subject to soil moisture availability. A third reason that one would expect glacier retreat ultimately to reduce basin runoff is that evaporation/condensation from snow or ice is typically low and often dominated by condensation, whereas an unglaciated surface would lose water by evaporation.

2. The scenarios represent glacier retreat followed by vegetation succession. However, retreat can also result in formation of lakes, which can accelerate glacier retreat and would ultimately provide an additional mechanism for reduced basin runoff via evaporation (Moyer et al., 2016). While it is likely not feasible to incorporate lakes into the model, this point should be acknowledged.

3. The model does not accommodate the development of a supra-glacial debris layer, which can reduce meltwater generation and the rate of glacier retreat. See Frans et al.

[Figure]

(2016). This point should at least be addressed as a discussion point if not incorporated into the model.

4. The analysis focuses on annual runoff, and the authors appropriately acknowledge the importance of considering seasonal runoff variations, particularly in late summer. This discussion could be extended by commenting on the relative magnitude of glacier contributions to seasonal and annual runoff (e.g., as a fraction of total runoff). Good references to draw upon are Frans et al. (2016) and Naz et al. (2014), both of which analyzed effects of glacier retreat on seasonal runoff.

5. The climate scenarios do not include decadal fluctuations, which can complicate peak water cycles – e.g., by generating transient periods of glacier advance, at least early in the warming phase. See, for example, Figure 4 in Clarke et al. (2015) and Figures 8 and 9 in Frans et al. (2016). Also, the magnitude of glacier runoff varies interannually, being greater in warm/dry years than in cool/wet years. See, for example, Naz et al. (2014). This compensating effect is an important aspect of glacier contributions to basin runoff that is not captured in the model.

6. The model scenarios are rather abstract, and I would encourage the authors to make a more structured effort to "map" the model scenarios into the real world. The authors should consider how they might synthesize their model results with results from the literature to develop a more nuanced conceptual model than those proposed by Jansson et al. (2003) and Moore et al. (2009).

7. Related to the preceding comment, the analysis does not consider the covariation of vegetation succession, climatic regime and elevation, or their influences on runoff generation. The authors cite only two papers to support the range of runoff ratios and three papers to support the parameterized model of landscape evolution. The authors should review a broader selection of papers to provide a better framing of their vegetation scenarios. A selection from the last five years includes Wietrzyk et al. (2018), Fickert et al. (2017), Whelan and Bach (2017), Eichel et al. (2015), Klaar et al.

[Figure]

(2015), Cowie et al. (2014) and Mizuno and Fujita (2014).

References

Clarke et al. (2015). Projected deglaciation of western Canada in the twenty-first century. Nature Geoscience 8: 372–377.

Cowie et al. (2014). Effects of glacial retreat on proglacial streams and riparian zones in the Coast and North Cascade Mountains. Earth Surface Processes and Landforms 39: 351–365, DOI: 10.1002/esp.3453.

Eichel et al. (2015). Conditions for feedbacks between geomorphic and vegetation dynamics on lateral moraine slopes: a biogeomorphic feedback window. Earth Surface Processes and Landforms 41: 406-419. DOI: 10.1002/esp.3859.

Fickert et al. (2017). Klebelsberg revisited: did primary succession of plants in glacier forelands a century ago differ from today? Alpine Biology 127: 17-29. DOI: 10.1007/s00035-016-0179-1

Frans et al. (2016). Implications of decadal to century scale glacio-hydrological change for water resources of the Hood River basin, OR, USA. Hydrological Processes 30: 4314-4329. DOI: 10.1002/hyp.10872

Klaar et al. (2015). Vegetation succession in deglaciated landscapes: implications for sediment and landscape stability. Earth Surface Processes and Landforms 40: 1088-1100. DOI: 10.1002/esp.3691

Mizuno and Fugita. (2014). Vegetation succession on Mt. Kenya in relation to glacial fluctuation and global warming. Journal of Vegetation Science 25: 559-570. DOI: 10.1111/jvs.12081

Moyer et al. (2016). Streamflow response to the rapid retreat of a lake-calving glacier. Hydrological Processes 30: 3650-3665. DOI: 10.1002/hyp.10890

Naz et al. (2014). Modeling the effect of glacier recession on streamflow response

using a coupled glacio-hydrological model. Hydrology and Earth System Sciences 18: 787–802

Whelan and Bach. (2017). Retreating glaciers, incipient soils, emerging forests: 100 years of landscape change on Mount Baker, Washington, USA. Annals of the American Association of Geographers 107: 336-349. DOI: 10.1080/24694452.2016.1235480

Wietrzyk et al. (2018). The relationships between soil chemical properties and vegetation succession in the aspect of changes of distance from the glacier forehead and time elapsed after glacier retreat in the Irenebreen foreland (NW Svalbard). Plant and Soil 428: 195-211

---

## Author Comment (AC1) · 29 Dec 2018

*B. Schaefli (Referee)*
*bettina.schaefli@unil.ch*

**General comments**

*This paper proposes to analyze the joint effect of glacier retreat and revegetation (due to climate warming) on the overall water balance of glacier-covered catchments for long term evolution (up to 500 years into the future). It does so with a simplified model whose possible outcomes are studied for different glacier retreat and revegetation scenarios, for two different climate types. The studied climates are continental and maritime climates, which are emulated by adjusting the glacier mass balance rate with elevation according to observed rates in these climates. No actual data is used in the presented study but the model parameters are selected in light of known /reasonable values for existing glacier catchments.*

*The idea of studying the possible evolution of catchment-scale water balance resulting from climate warming with a simplified model is appealing; it has the potential to explain in simple terms the possible outcomes (temporal increase of total basin runoff, overall decrease on the long run) without obscuring the involved mechanisms by a complex input-output model. In its current form, the results of the analysis are however hardly surprising and essentially say that "with more vegetation we get less runoff", which corresponds to an oversimplification of high alpine hydrology.*

Thank you for your comments and careful review. Although some of the results may not be surprising, we feel that previous literature has not systematically analyzed the parameters influencing annual runoff, and thus our results provide a simple framework for understanding variations in runoff that should be relevant to a broad range of researchers and resource managers. As pointed out in this review and in the other reviews, additional complexity could be added to the model that would produce positive and/or negative feedbacks, but that would not change the general results from this study. In the introduction, we highlight that we are trying to understand (1) how a suite of fundamental parameters (glacier slope, climate regime, etc.) control the shape of the long-term annual runoff curves and (2) how sensitive the runoff curves are to changes in these parameters.

*I am a hydrologist by training, with little knowledge in ice flow modelling. From my perspective, the used one-dimensional, depth- and width-integrated flow model, combined with different glacier mass balance rates seems to be a reasonable approach to generate different glacier retreat scenarios under climate warming. I find it, however, surprising that the authors choose an approach that does not allow to study the effect of the actual glacier shape (here a simple rectangle has been chosen) and that this aspect is not further discussed.*

The rate of glacier volume change, which drives variations in glacier runoff over secular time-scales, is governed by two feedbacks: a negative feedback with glacier length and a positive feedback with glacier surface elevation. These feedbacks are well captured by

simple flow line models, although it is correct that spatial variations in glacier width will modify the glacier evolution. We tested the model sensitivity to glacier width by using a trapezoidal basin (in map view) whose width varies from the ice divide to the terminus by ± 5 degrees. These variations have limited effect on peak runoff (~1%) and changes in time to peak and end runoff were easy to predict. Essentially, all other things equal, glaciers with large accumulation areas have higher end runoffs due to the smaller fractional area change and a slower decrease to end runoff. The large accumulation areas provide some buffer against climate warming as long as the glacier does not fully disappear. In the manuscript we now motivate our choice of using a parallel-sided valley and discuss how variable glacier width might affect the variations in runoff.

*Regarding the hydrological side of the study, I have to admit that as I hydrologist I can only warn against the use of such oversimplified assumptions without sufficient discussion of the implications. To actually study the fundamental controls on the high alpine water balance, these fundamental controls and what we know thereof should be reviewed in detail before building a model.*

*My critic is the following: The parameterization of the effect of colonization is summarized by* **two simple assumptions***: "First, we assume that the catchment becomes increasingly vegetated following deglaciation and that the type of vegetation only depends on time since deglaciation. Second, as areas of the catchment become colonized, the rate at which water is evapotranspired increases until reaching a maximum value representative of the climax vegetation state."* **While the first assumption seems reasonable** *(some references would certainly be useful), ...*

The first assumption is based on the time since deglaciation being highly correlated with vegetation types, biomass, and cover (Crocker and Major, 1955; Burga et al., 2010; Chapin et al., 1994; Klaar et al., 2015; Whelan and Bach, 2017; Fickert et al., 2017; Wietrzyk et al., 2018). The assumption does not include any variations in vegetation regrowth with altitude, which have been shown to affect vegetation growth rates primarily through its influence on air temperature (Cowie et al., 2014; Whelan and Bach, 2017). Yet, succession rates have been shown to be comparable at different altitudes throughout glacier recession as changes in air temperature with altitude are offset by climate warming (Fickert et al., 2017). We have added the following citations to the manuscript, and thank you for the suggestion.

Crocker, R. L. and Major, J.: Soil Development in Relation to Vegetation and Surface Age at Glacier Bay, Alaska, J. Ecol., 43, 427–448, 1955.

Burga, C. A., Krüsi, B., Egli, M., Wernli, M., Elsener, S., Ziefle, M., Fischer, T., and Mavris, C.: Plant succession and soil development on the foreland of the Morteratsch glacier (Pontresina, Switzerland): Straight forward or chaotic?, Flora, 205, 561–576, https://doi.org/10.1016/j.flora.2009.10.001, 2010.

Chapin, F. S., Walker, L. R., Fastie, C. L., and Sharman, L. C.: Mechanisms of Primary Succession Following Deglaciation at Glacier Bay, Alaska, Ecological Monographs, 64, 149–175, https://doi.org/10.2307/2937039, https://esajournals.onlinelibrary.wiley.com/doi/abs/ 10.2307/2937039, 1994.

Klaar, M. J., Kidd, C., Malone, E., Bartlett, R., Pinay, G., Chapin, F. S., and Milner, A.: Vegetation succession in deglaciated landscapes: implications for sediment and landscape stability, Earth Surface Processes and Landforms, 40, 1088–1100, https://doi.org/10.1002/esp.3691, 2015.

Whelan, P. and Bach, A. J.: Retreating Glaciers, Incipient Soils, Emerging Forests: 100 Years of Landscape Change on Mount Baker, Washington, USA, Annals of the American Association of Geographers, 107, 336–349, https://doi.org/10.1080/24694452.2016.1235480, 2017.

Fickert, T., Grüninger, F., and Damm, B.: Klebelsberg revisited: did primary succession of plants in glacier forelands a century ago differ from today?, Alpine Botany, 127, 17–29, https://doi.org/10.1007/s00035-016-0179-1, 2017.

Wietrzyk, P., Rola, K., Osyczka, P., Nicia, P., Szymański, W., and We grzyn, M.: The relationships between soil chemical properties and vegetation succession in the aspect of changes of distance from the glacier forehead and time elapsed after glacier retreat in the Irenebreen foreland (NW Svalbard), Plant and Soil, 428, 195–211, https://doi.org/10.1007/ s11104-018-3660-3, 2018.

Cowie, N. M., Moore, R. D., and Hassan, M. A.: Effects of glacial retreat on proglacial streams and riparian zones in the Coast and North Cascade Mountains, Earth Surface Processes and Landforms, 39, 351–365, https://doi.org/10.1002/esp.3453, 2014.

*... the second assumption omits an important body of hydrological literature of the effect of vegetation on the water balance, and in particular the effect of forest (e.g. Andreassian, 2004). Forests show typically increased ET fluxes during younger states as compared to the climax state.*

*Whether the typical vegetation succession to be expected in glacier catchments leads to a continuous ET increase with vegetation cover increase, remains to be demonstrated. I am not aware of literature on this topic (but it might well exist of course). In general the evolution of hydrological / geomorphological / pedological processes in moraines (and related runoff processes) can be assumed to be still largely unknown (see an ongoing project description here: http://gepris.dfg.de/gepris/projekt/318089487?language=en).*

The second assumption is that as areas of the catchment become colonized and vegetation biomass increases, the amount of precipitation that does not contribute to runoff on an annual scale, ET, increases until reaching a maximum value representative of the climax vegetation state. The assumption is based on a general understanding of

the relationship between biomass, vegetation cover and decreased basin runoff. A variety of processes are expected to cause annual ET to increase including: increases in vegetation biomass, type, percentage cover, and temperature (Jaramillo et al., 2018; Andréassian, 2004; Barnett et al., 2005), yet as the reviewer rightly points out there are few studies on changes in evapotranspiration throughout vegetation succession following deglaciation. Results for non-glaciated paired watershed studies show a clear decrease in annual basin runoff moving from the time of initial reforestation to the establishment of climax forest (Andréassian, 2004; Filoso et al., 2017). Changes in evapotranspiration rates through the transition period from initial reforestation to climax state in non-glaciated basins is variable. Some studies show approximately monotonic decreases in annual basin runoff from reforestation to climax forest (Andréassian, 2004, and references within; see there Fig. 8). However, others show a non-linear decrease in basin runoff after deforestation, with younger states having higher evapotranspiration rates than climax state (Andréassian (2004), and references within; see there Fig. 9). Thus, there are two scenarios and the debate between them continues, either evapotranspiration on newly revegetated land is lowest at first and progressively increases until climax state or evapotranspiration is initially lowest and increases rapidly before decreasing and stabilizing above deforestation levels at climax state. These conflicting results have been explained as particular to different species of tree with the latter, non-linear increase in evapotranspiration, measured primarily for eucalyptus trees (Andréassian, 2004).

Our modeling is of plant growth in a previously deglaciated basin, where transitions in evapotranspiration have yet to be extensively studied. However, based on evidence for the first assumption we can assume that vegetation biomass, types, and cover are all increasing with time since deglaciation. There are multiple studies showing that increased biomass and reforestation leads to higher levels of evapotranspiration and decreased annual basin runoff (Sun et al., 2010; Klaar et al., 2015; Jaramillo et al., 2018; Bosch and Hewlett, 1982; Andréassian, 2004). In our general modeling we choose to model evapotranspiration as monotonically increasing in a stepwise manner throughout vegetation growth for the following reasons. First, we are attempting to study general basin characteristics so exceptions to general rules (e.g., eucalyptus trees) are of less importance. Second, the step wise increase in ET allows us to focus on specific stages of vegetation and not the exact transition between stages which is less well understood; most studies show an eventual increase in ET and interception after vegetation reaches a climax state (Andréassian, 2004). These two assumptions provide the basis for our landscape modeling throughout glacier recession. We have more clearly delineated the justification for these assumptions in the methods.

Jaramillo, F., Cory, N., Arheimer, B., Laudon, H., van der Velde, Y., Hasper, T. B., Teutschbein, C., and Uddling, J.: Dominant effect of increasing forest biomass on evapotranspiration: interpretations of movement in Budyko space, Hydrology and Earth System Sciences, 22, 567–580, https://doi.org/10.5194/hess-22-567-2018, 2018.

Andréassian, V.: Waters and forests: from historical controversy to scientific debate, Journal of Hydrology, 291, 1 – 27, https://doi.org/https://doi.org/10.1016/j.jhydrol.2003.12.015, 2004.

Barnett, T. P., Adam, J. C., and Lettenmaier, D. P.: Potential impacts of a warming climate on water availability in snow-dominated regions, Nature, 438, 303 EP –, https://doi.org/10.1038/nature04141, 2005.

Sun, G., Noormets, A., Gavazzi, M., McNulty, S., Chen, J., Domec, J.-C., King, J., Amatya, D., and Skaggs, R.: Energy and water balance of two contrasting loblolly pine plantations on the lower coastal plain of North Carolina, USA, Forest Ecology and Management, 259, 1299 – 1310, https://doi.org/https://doi.org/10.1016/j.foreco.2009.09.016, 2010.

Bosch, J. and Hewlett, J.: A review of catchment experiments to determine the effect of vegetation changes on water yield and evapotranspiration, Journal of Hydrology, 55, 3 – 23, https://doi.org/https://doi.org/10.1016/0022-1694(82)90117-2, 1982.

*I do think that the approach is interesting. The hydrological assumptions should however be a bit more elaborate, including good references for glacier catchments and a detailed review of what we know today about the evolution of the water balance of newly vegetated areas in such catchments. If no sufficient literature can be found, possible hypotheses should be discussed in detail. This literature review should also include the important ongoing discussion what the effect of decreases in snow to rainfall ratios has on the catchment-scale water balance (Berghuijs et al., 2014). The relative decrease of snowfall might significantly contribute to the reduce of basin-scale runoff (add to the effect of vegetation).*

We have found equivocal studies on the impact of changes from snow to rainfall on annual streamflow. Basins in southeast Alaska show strong seasonal changes but no discernible trend in annual streamflow from moving from snow-dominated to rain-dominated climate regimes over 20 year climate oscillations (Neal et al., 2002). A review of studies in the western United States found no clear trend in how mean annual streamflow responded to changes in precipitation phase across different basins (Tague and Dugger, 2010). Finally, a study of non-glaciated basins across North America suggests that a change in phase of precipitation from snow to rainfall results in larger interannual variability, and lower mean annual streamflow (Berghuijs et al., 2014). These differing results do not allow for the determination of a simple modeling parameter to include for changes in annual runoff associated with changing precipitation regime in a glaciated basin. We briefly justify why we choose to not include the effect of changing from snow to rain in our modelling of annual runoff for glaciated basins. We also mention the possible effect of the alternative hypothesis and how it affects our results.

Neal, E., Walter, M. T., and Coffeen, C.: Linking the pacific decadal oscillation to seasonal stream discharge patterns in Southeast Alaska, Journal of Hydrology, 263, 188 – 197, https://doi.org/https://doi.org/10.1016/S0022-1694(02)00058-6, 2002.

Tague, C. and Dugger, A. L.: Ecohydrology and Climate Change in the Mountains of the Western USA – A Review of Research and Opportunities, Geography Compass, 4, 1648–1663, https://doi.org/10.1111/j.1749-8198.2010.00400.x, 2010.

Berghuijs, W. R., Woods, R. A., and Hrachowitz, M.: A precipitation shift from snow towards rain leads to a decrease in streamflow Nature Climate Change, 4, 583–586, 2014.

*Similarly, a topic that should be discussed (even if not included in the analysis) is the interaction between glacier retreat and groundwater recharge. Not much is known so far about this topic but glacier retreat might change the relative amount of water that is available to vegetation in the non-glaciated part.*

Thank you for this suggestion. Reviewer 3 also pointed out a number of processes that we neglected in our model. In response to those comments, we briefly discuss how processes such as groundwater recharge might modify our model results for annual basin runoff. We note that changes in glacier mass balance have been shown to affect groundwater recharge, however the impact to basin runoff is seen more strongly at seasonal rather than the annual timescales we are modeling (e.g., Liljedahl et al., 2017).

Liljedahl, A.K., A. Gädeke, S. O'Neel, T.A. Gatesman, and T.A. Douglas (2017), Glacierized headwater streams as aquifer recharge corridors, subarctic Alaska, *Geophys. Res. Lett.*, 44, 6875-6885, doi:10.1002/2017GL07383.

*To summarize, to increase the value of this study, I suggest a good literature review of the impact of glacier retreat and the associated reduction of snow- to rainfall ratio on the water balance of high alpine catchments. Based on this, key processes and their synergy and possible unknowns should be identified. Based on this, the hydrological model can either be kept as is (but with more realistic future scenarios) or be refined. At the very least, the hydrological simplifications should be more explicitly discussed.*

We acknowledge that there are limitations to the assumptions that we made for both the landscape and glacier models. These assumptions were made due to either a desire to understand a few fundamental parameters that influence basin runoff or a lack of consensus on various processes. We prefer not to add additional model complexity at this point and chose to focus on the key processes/parameters. In the revised manuscript we added justification for our chosen model parameters and also discuss how some of the parameters identified by the reviewer that were not included in our model, such as the snowfall to rainfall ratio and changes in ET, may affect trends in runoff (see also response to reviewer #3).

***Detail comments:***

- *Regarding the future ET fluxes, the reference to a paper that studied forest versus crop / pasture across the globe in non-mountain environments (Zhang et al., 2001) is probably not adequate.*

This issue was also raised by reviewer #3, who suggested a number of additional studies that we have now included in the paper.

- *The concept of **"runoff ratio"** is an engineering concept that was developed to separate precipitation into surface runoff and infiltration at the event scale (e.g. for the application of the so-called rational formula). What is used in this model is the **"annual runoff ratio"**, which is the ratio between total basin runoff and the total incoming precipitation. The total basin runoff is the sum of direct surface runoff and fast and slow subsurface runoff processes (and not the "runoff over an area of land"; the latter are the result of soil – vegetation interactions and groundwater recharge / release processes. This should be clear to avoid confusion for non- hydrologists.*

Thanks. We have clarified this in our revision.

- *the conclusion should give clear indications about what should be explored on the hydrological side (not just the glaciological side)*

We agree, and we have addressed model limitations in more detail in the manuscript (discussed above and in response to reviewer #3).

---

## Author Comment (AC2) · 29 Dec 2018

**Summary**
*In this study the basin (glacier) peak water trajectory, following glacier retreat, is modelled using a glacier flow model in combination with some parameterizations, to simulate glacier retreat and changing vegetation in the non-glacierized areas of the basin. The effects of basin slope, climate type (maritime and continental), vegetation rate and type, and climate change scenario (RCP2.6 RCP8.5) on this trajectory are tested. The results show that slope and climate type influence the magnitude and timing of peak water, and this is related to the glacier response time. A continental climate and shallow slopes cause a higher increase in basin runoff and a later time of peak runoff, compared to a maritime climate and steep basin slopes. The effect is more pronounced in the RCP8.5 scenario. Vegetation rate and type is influencing how fast runoff levels decrease after peak water to pre- peak water runoff levels and vegetation type determines how much runoff drops after peak runoff compared to initial runoff levels.*

*The modelling approach is rather mathematical, in contrast to many other glacio-hydrological studies published in HESS. This allows to perform an interesting sensitivity study, which is of interest for the HESS community. However, the more glaciological way of describing a glacierized hydrological system as presented in this study, requires more clarity, explanation and discussion when publishing in a hydrology journal (HESS). Please find my explanation, together with some other concerns below. Apart from that, the manuscript is generally well written and the figures are nicely presented.*

Thank you for this feedback. Our goal with this paper was to write it in a way that would be of interest to both glaciologists and hydrologists, and we have therefore made a concerted effort to revise the model description to make it more accessible to non-glaciologists.

**Major issues**
*1. Modelling framework*
*The study uses a simple glacier flow model in combination with parameterizations of runoff ratios to model vegetation succession in the non-glacierized parts of the basins. Together with some climate "input", this is coupled to calculate basin runoff, glacierized runoff and nonglacier runoff over time. However, the problem is that the description of the model in the different equations and sections is not well connected (e.g. how the modelling of glacier dynamics is connected to the calculation of Qg or in which equations parameters are changing (apart from C and T)). This is important to better understand and interpret the results.*

*Equation 1 gives a good overview of the main modelling framework. However, from the other equations given in the methods section it is not always clear how they fit the calculation of the total basin runoff. The description of the precipitation input is sometimes a bit confusing. Why is it a separate section? And why is there written that it includes the solid and liquid fluxes? This is a bit confusing since there is no temperature input involved.*

*Maybe it should be also made clear that precipitation "input" is constant every year. Precipitation is in this study also not a real input to e.g. the glacier, because the mass balance is another parameter partly independent of precipitation. Please clarify the sentence "precipitation at sea level is chosen to ensure that the precipitation at elevation exceeds glacier accumulation rates". Does it mean that precipitation should fit the mass balance rates above zero? And what is the exceeding precipitation assumed to be? Can this be indicated in Figure 1?*

In response to these questions:
- The precipitation parameterization is input as a separate section because it is needed for both the glacier and non-glacier components of the model (in the subsequent sections, which we now refer to when presenting the precipitation parameterization).
- The precipitation parameterization describes the total precipitation flux, and therefore includes both solid and liquid precipitation. This is important to note because snow that falls in winter melts and runs off in the summer and therefore contributes to the annual basin runoff. The exception is in the glacier accumulation area, where not all of the snow will melt during the summer. This is accounted for through the mass balance parameterization; in the accumulation area, the mass balance rate is positive, and therefore the amount of runoff generated at a specific location is less than the precipitation flux. Lower on the glacier, the mass balance is negative, and thus the runoff produced exceeds the precipitation flux. We have added a few sentences to the methods to clarify why the precipitation flux includes both solid and liquid precipitation and to indicate that our method for calculating glacier runoff is identical to calculating the sum of rain plus glacier melt (without the need of a climate model that calculates those terms independently).
- We require the precipitation at elevation to exceed the glacier mass balance rate because the amount of snow that accumulates on the glacier can't be more than the amount of snow that falls on the glacier. If the precipitation rate equals the mass balance rate then there is no summer melt; if the precipitation rate exceeds the mass balance rate then there is some summer melt. We have slightly reworded this sentence.

*In the section about the glacier runoff and the glacier model, it might be more clear when the section starts with the description of the glacier model and then show that the output of this glacier model (surface area of the glacier $\Omega$ g) is used to calculate the glacier part of the total basin runoff (and how it influences $\Omega$ n in the next section). Why is P(z) written in equation 2, but is "(z)" left out in equation 4? What does "min" indicate in equation 4? And why is there a maximum mass balance (Bmax)? Why can P increase with height but B not? What is meant with glacier hypsometry (L21 P4)? If it refers to equation 3 it only indicates length changes (since the width is constant), or does it also include the glacier thickness due to "z" in P and B? What does small h mean in equation 5? Is this the slope? What is solved from equation 5? And how does it relate to equation 7? I think some more explanation here would be beneficial.*

We prefer to start this section with the equation describing glacier runoff (equation 3) because for non-glaciologists this is the only equation that really matters. Everything that follows are basically details about the glacier model. To help clarify, though, we have added a statement that indicates that equation 3 changes with each time step because of changes in glacier surface elevation and extent, which we account for with a glacier flow model.

Additional comments:
- "(z)" should be included in equation (4); thanks for catching this.
- "min" refers to the minimum of two numbers i.e. the balance rate increases with elevation until reaching a maximum value, B_max.
- The mass balance profiles, which reach a maximum value at high elevations, are based on field observations from many glaciers. We have now added a reference to Van Beusekom et al. (2010) that demonstrates this phenomenon for both maritime and continental glaciers. The leveling off of the balance rate at high elevations is probably due to several processes, including things like refreezing of meltwater that percolates into firn and the length of the melt season, but it can also be understood in terms of changes in precipitation type (solid vs. liquid) with elevation. At high elevations precipitation occurs mainly as snow, and since ablation rates scale with temperature (and therefore elevation), the difference between accumulation and ablation is linear --- and observations suggest that the difference between these two is roughly constant at high elevations. At low elevations, a larger fraction of the precipitation falls as rain and does not contribute to the glacier's mass balance; thus the mass balance profile is more strongly affected by ablation processes there, resulting in a bending of (or kink in) the mass balance profile.
- We replaced glacier hypsometry with "glacier geometry (surface elevation and extent)".
- Little 'h' is the surface elevation and is now indicated as such.
- Equation 5 is solved for the velocity, which is then used to calculate changes thickness via Equation 7. This has been clarified.

Van Beusekom, A. E., O'Neel, S. R., March, R. S., Sass, L. C., and Cox, L. H.: Re-analysis of Alaska benchmark glacier mass-balance data using the index method, USGS Scientific Investigations Report 2010-5247, 16 p., 2010.

*The "t=0" in L1 P6 is a bit confusing with the later explanations that the climate in the model is kept constant during the first 10 years. What is t(0) in this case? Start of the simulations or when a portion of the catchment is deglaciated? Related to that it is also confusing that it is written that the climate is kept constant (for each climate type?) to reach a steady state (spin-up) and is then changed by changing the ELA, but then the climate is held constant for 10 years (no change in ELA)? Please reorder. Also the definition of "constant climate" (L25 P6) only becomes clear later in the text when it is explained that climate change is modelled by changing the ELA. The last sentence of the methods also requires some more explanation, that the simulations continue until the glacier have reached a new steady-state. How can the glacier reach a steady state when*

*the ELA is increasing every timestep, especially in the RCP8.5 scenario? Is there a maximum ELA?*

We have changed our graphs so that t=0 is the time that the climate starts to change, and we removed the text about holding the climate constant for the first 10 years of each simulation.

The last sentence states that the simulations are run until the glaciers reach a new steady-state *or completely disappear*, the latter of which happens in the RCP8.5 scenarios.

*Apart from the methods model description also other parts of the manuscript sometimes lack clarity:*
*• It would help if the key metrics described in the results are indicated in a conceptual figure. Especially the time to pre-retreat basin runoff and end basin runoff would get more clear from such a graph*

We agree and think that this is an excellent suggestion. Thanks. We have added a figure and a statement in the introduction about the metrics that we are assessing.

*The "Thus" sentences in the manuscript are not always straightforward:*

- *– "thus the basins do not have the same length" (L8 P7) – this depends on climate type (and thus mass balance gradient) but also on slope? It would help if the initial glacier areas/lengths and volumes for all simulations (climate type and slopes) are given, together with their change over time. In that case the fractional volume changes e.g. for steep glaciers and the different climate types could be better interpreted. Why is for example the fractional volume and area change similar for both climate types, but the peak runoff differently – due to a larger volume in the continental climate? It also helps to visualize that there is a limited amount of newly vegetated land at peak runoff. It would be good to indicate/explain differences in glacier geometry due to climate type and slope in the results or methods based on the equations, e.g. why shallow sloped basins contain longer glaciers.*

This sentence has been re-worded.

- *– "Thus the model results tend to overemphasize the relative importance of glacier runoff on basin runoff" – because in reality one does not start with 100*

This sentence has been re-worded.

- *– "Thus we assume that the basal shear stress is at the yield stress" – please explain the "thus"*

"Thus" has been replaced with "In other words".

*2. Clarity*

- *In the results section: why are results sometimes explained for one of the two climate scenarios only?*

  This is done when the results that are being described are universal, or in other words, that trends are the same for both climate scenarios.

- *What is meant with glacier geometries? Slope, length, thickness?*

  Glacier geometry was replaced with bed slope.

- *What is the reason that glacier runoff peaks before basin runoff? The decreases in precipitation on glaciated land also influence basin runoff?*

  Peak basin runoff lags peak glacier runoff because nonglacier runoff continues to increase when glacier runoff is at a peak. This can be understood through a simple analysis of the basin runoff given in Equation 1:

  $Q\_s = Q\_g + Q\_n$

  Taking the time derivatives of both sides:

  $dQ\_s/dt = dQ\_g/dt + dQ\_n/dt$

  When the glacier runoff is at a peak, $dQ\_g/dt = 0$ and therefore $dQ\_s/dt = dQ\_n/dt$. Because the nonglacier runoff is increasing at this time as new bedrock is being exposed, the basin runoff must also be increasing, which implies that it has a later peak. We have added a paragraph to the end of the discussion which explains this and, in addition, explains the observation that peak basin runoff exceeds peak glacier runoff (in both absolute and relative terms).

- *What is magnitude in case of end basin runoff? The magnitude is smallest for peak basin runoff, but largest for end basin runoff in case of a heavily forested state? (P8).*

  The magnitude of end basin runoff is the amount of basin runoff that occurs when the glacier reaches a new steady state (RCP2.6) or disappears (RCP8.5). P8 L13-14 "Overall… ...(low runoff ratio)." states that the magnitude of end basin runoff is *smallest* when the vegetation progresses to a heavily forested state.

*3. Structure*
*The introduction section of this manuscript lacks the description of a clear knowledge gap. It should be emphasized more what is new about this study (landscape coupling?) and what we do not yet know. The results section includes quite some interpretation, and even refers to the discussion (glacier response times). The results section also includes text about key metrics that should shift to methods.*

We have added a conceptual figure, and associated text, to the introduction to clarify the knowledge gaps and describe what is new in this study. This also allowed us to introduce

the metrics that describe the changes in runoff curves. Nonetheless, we do feel that our initial draft did describe the knowledge gaps that we are addressing, particularly in paragraph 4 of the introduction. For example, we wrote "..., these case studies do not elucidate the broader geomorphological and glaciological controls that govern the hydrological responses of watersheds to ongoing glacier recession."

*4. Discussion and implication*
*In the discussion the hydrological changes (changes in annual runoff) are discussed together with their controls and compared to other literature. However, the implication of the quantitative analysis (as presented in the introduction) is lacking. What do the numbers mean and how can they be transferred to glacierized catchments around the world? Some numbers are compared, but it is not always clear which part of the graphs (trajectory) agree with observations. The simulations all start with 100% glacier cover, but what can we learn from that when a catchment has e.g. 50% glacier cover? Will it have the same variations? And what if the glacier hypsometry has not a fixed width? Why has a 1D model been chosen? Has t(0) been in the past for glacierized catchments and can we expect a similar peak runoff and rate of decline in annual runoff? Is e.g. the size of the glacier modelled in this study representative? Other aspects that could be more emphasized is the drop of annual runoff below pre-retreat levels, which is e.g. not found/modelled in other studies (e.g. Huss Hock, 2018). Also the importance of including vegetation could be more stressed and compared with other studies (where it is often neglected).*

Some specific replies to these questions:
- Our goal with this study is not to describe the specific responses of particular glaciers or regions, but rather to develop a theoretical understanding of how variations in annual basin runoff depend on several key parameters. From our study a reader should be able to make an educated guess about how basin runoff will vary for their glacier of interest. More accurate, glacier specific predictions would require designing a coupled glacier-landscape model for a particular region.
- More detailed comparisons between model results and observations are difficult/impossible owing to a lack of streamflow measurements over decadal-to-centennial time scales.
- The impact of initial glacier coverage on the results was initially explored but had the straightforward effect of adding a constant (the basin runoff from a portion of the basin with climax vegetation) to any calculation of basin runoff. Thus, having a smaller initial glacier coverage reduces the impact of glacier loss on basin runoff in an easily predictable way, which we now discuss in the manuscript.
- See response to reviewer #1's comments regarding the impact of glacier width and the choice of using a 1D model.
- The question of dis(equilibrium) is an interesting one, as glaciers are probably never truly in a steady-state, and the distance from steady-state may have interesting consequences for interannual variability in runoff. We now discuss this in more detail in the conclusions and leave it for future work.
- The drop in annual runoff below preretreat levels is not found in other studies that do not account for vegetation. We emphasize this point in the manuscript.

*Also the glacier response time is discussed, as an explanation why slope and climate type influence the hydrological response. Why is peak basin runoff related to the time a glacier needs to responds to climate change? This would only be half way (the time it needs to reach a new steady state)? Can the different simulations for which a response time is calculated also be indicated in Figure 8? The conclusions that are drawn in the text can now not be seen in the Figure. Is the response time – peak runoff relation also influenced because the ELA increases every time step?*

Peak basin runoff occurs relatively early during glacier recession, when glacier runoff is a large proportion of total runoff (see Figure 7) and is therefore a dominate control on total runoff. Peak glacier runoff is related to the glacier response time because glaciers with long response times are pushed farther out of equilibrium and take longer to evolve back toward a steady state.

The glacier response times do not vary with any changes in the landscape parameters, and therefore only the peak runoff and time to peak runoff are affected by climate change and vegetation types/rates (in other words, the vertical axes in Figure 8 are affected by vegetation and climate change but the horizontal axes are not). We now clarify what data we are plotting in this figure. Use of a different climate change scenario (e.g., RCP2.6) would change the curves, with slower rates of climate change causing smaller fluctuations in basin runoff (see previous figures).

**Specific remarks**
*L7 P1: "rate of climate change" – what does rate mean here? Scenario might be more clear*
Changed to "climate change scenario".

*P1 abstract: "Peak basin runoff" – use magnitude of peak basin runoff as in rest of paper to be more clear*
Added "magnitude" to improve clarity.

*L24 P1: "Moreover, changes in runoff...ecological function of downstream aquatic ecosystems" – The order of the sentences is strange here, because one first reads that changes in glacier runoff only effect the downstream aquatic ecosystems, but on the next page it is described how all the ecosystem services will be affected by changing glacier runoff.*
The changes to the structure and function of aquatic ecosystems are an example of how changes in glacier runoff propagate downstream that is separate from the ecosystem services listed previously. We left the order of the sentences as they were originally written.

*L2 P2: "Glacier runoff. . ..water budget" – this sentence does not fit here, move up or connect better*
We added some text to improve the connection to the previous sentence.

*L5 P2: "lower baseline" – only Moore et al. (2009) show a lower baseline, Jansson et al. (2003) not. Also Huss Hock (2018), for example, show no lower baseline. So either explain why there is a lower baseline, or leave it out in the introduction and discuss the differences presented in the literature in the discussion (or discuss in the introduction)*

Both the Jansson et al. and Moore et al. papers (cited in this sentence) show a lower baseline. And, while it is true that the Huss and Hock paper (not cited in this sentence) did not show a lower baseline, they explicitly acknowledged that they "do not consider other processes in the gradually growing deglacierized proglacial area, such as evapotranspiration or changes in groundwater recharge and land cover" that are responsible for the lower baseline in annual runoff seen in our study. Thus we left the sentence as written.

*L7 P2: "increase roughly 50 percent by end of century" – compared to what?*

We changed this to "during the 21st century" to be more precise.

*L11 P2: "On a global scale. . ..South America" – be more explicit here, Arctic, Canada and Russia have higher glacier coverage basins? In Asia, Europe and South America glaciers have retreated and therefore lower glacier coverage?*

We added the phrase "where glacier coverage is lower" after "Asia, Europe, and South America".

*L14 P2: How can "Stahl and Moore (2006)" be both cited as a study on individual catchments and on regions? Nolin et al. (2010) is a study on a specific catchment so why mentioned as a study focused on the regional scale? Huss and Hock (2018) is a global scale study? "case studies" in the next sentence does not fit all of the references mentioned here.*

Stahl and Moore 2006 is listed as both an individual catchment and region because it reports data on runoff change in over 100 individual glacierized catchments and uses these results to draw conclusions about changes in glacier runoff across British Columbia, which we consider to be a region. The Nolin reference was misplaced and has been moved earlier in the sentence. This is getting super particular but the Huss and Hock paper models changes in glacier runoff for 56 large basins, which is not all of the glacierized basins world. All but one of of the basins modeled by Huss and Hock are categorized into 4 regions in the paper: Asia, Europe, N. America and S. America. Thus, the paper provides insight into future glacier runoff change in these regions and is appropriate as referenced.

*L18 P2: what does "also" mean here?, same for "also" in line 21?*

On line 18, we replaced "also" with "in addition". We did not replace the "also" in line 21 because we feel the meaning should be self-evident to the vast majority of readers.

*L21 P2: reduce the use of "the fact that" throughout the manuscript*

This phrase was used 5 times in 18+ pages of text. We reduced our use of the phrase by 40%.

*L25 P2: "annual basin runoff" is used mostly in the paper, but in title and introduction "water yield" is used – why?*
We now use "annual basin runoff" throughout the paper.

*L1 P3: "definition 5" – please explain*
We have added "total runoff from the glacier surface" to the parenthetical remark, which is the 5th definition for glacier runoff presented in O'Neel et al., 2014.

*L4 P4: notation of variables with an overdot to indicate width average – is overdot usually not used to indicate a derivative?*
We use the dot indication to denote a rate, or derivative with time, they just happen to also be width averaged.

*L7 P3: "precipitation at elevation" – which elevation?*
Assuming you mean P4. Changed to "Precipitation at *any* elevation", but we are specifically talking about precipitation in the accumulation area.

*L8 P5: "timestep" – indicate that timestep is one year*
The time step is .08 of a year and this has been clarified.

*L15 P5: "runoff ratio (the ratio of precipitation to runoff over an area of land)" – switch precipitation and runoff -> the ratio of runoff to precipitation*
Change has been made.

*L24 P5: "runoff ratios range from 0.5 (forest) to close to 1 (ice)" – on the next page it is written that runoff ratios are 1, and that it represents rocky high elevation environment with no vegetation?*
We changed the sentence to "...~1 (ice or rocky alpine terrain with no vegetation)".

*Eq. 11 and 12 P6 and P7: indicate (e.g. as subscript) that equation is for RCP2.6 and the other for RCP8.5*
We added the requested subscripts to the equations.

*L8 P7: "As the glacier recedes", add comma*
Comma added.

*L3 P9: "on slope and climate type and is related to the glacier response" – remove "and" or is another variable forgotten here?*
Removed "and"

*L10 P9: Fig 5a,b – this should be figure 5 a and c – see also other references to Figure 5 in this part of the results*
Good catch, the corrections were made.

*L9 P11: "slightly longer times" – longer times of what?*
Added "to peak and preretreat basin runoff" to clarify.

*L5 P12: "for all glacier geometries"- what is meant here? Slope?*
Changed "geometries" to "bed slopes".

*L5 P13: "final steady state basin runoff following glacial recession is strongly influenced by the rate and type of vegetation" – do you mean here the final steady state basin runoff or also the timing of the end basin runoff? In the first case, this sentence contradicts the results*
We clarified that steady state basin runoff following glacial recession is strongly influenced by the type of vegetation that colonizes ice free areas of the catchment.

*L15 P13: "longer response time" – what is response time here?*
Added "to peak runoff" to clarify the reference to response time.

*L29 P13: "end glacier runoff" – what is end glacier runoff?*
Deleted "end" so that the term "glacier runoff" is consistent with the terminology in Nolin et al. (2010)

*Figures:*

- *Fig.                                                                                         1:*
  *– Can you indicate ELA in fig. 1c?*
  *– For clarity it might help to also plot the lines for a maritime climate and if possible also for the RCP2.6 scenario*

We now indicate in the caption that the ELA occurs where the balance rate is 0. We prefer to only plot one climate type and RCP scenario to keep the figure clean.

- *It would be helpful to have the same x and y axes in all figures, since for the interpretation of some results one needs to look at several graphs*

We have changed our figures to have similar x- and y-scales when presenting basin runoff curves.

- *Why is legend in some figures in the right graph and in others in the left graph?*

We have now moved the legends to the left panels.

- *Please indicate the degree symbol in the "slope" legends*

We have made this change.

- *When looking at the figures it is not directly clear what is compared in the left and right graphs, although it is indicated in the figure captions. Could the figures get a title or a label in the graph so it is clear what is compared in both?*

Adding a title would be redundant with the caption, so we have left the figures as is.

- *Fig. 2 :*

- o    – *Why is y axis starting at 0, but at 70 in figure 4?*
- o    – *What determines the length of the (horizontal) line indicating after peak runoff in figure a? I assume glaciers have disappeared and since no vegetation is present in figure 1 no final vegetation state needs to be reached*

We have changed the figures to have the same scales. See above comment.

The horizontal lines arise because the model is required to run through the full vegetation succession, even though the runoff ratio doesn't change.

- • *Fig. 3 and 4 and 5: why is the basin slope 5 and does figure 2 not show a slope of 5 degrees?*

Figure 2 shows the range of runoff curves, and the curve for slope 5 can be inferred from the curves that are presented in Figure 2.

- • *Fig. 5: Missing in caption, results are only shown for maritime climate?*

The caption does indicate that the results are for a maritime climate.

- • *Fig. 6:*
  - o    – *add symbols as legend*
  - o    – *What determines the end of the simulation in both graphs? Compared to Figure 2a the results stop earlier in Fig. 6a. Also for 6b this is not clear*

We use the legend to describe the color of the curves. The meaning of the symbols are indicated in the caption.

In Figures 2a and 6a, the glaciers disappear at the same time (for example, at t=300 years for the dark blue curves). The extra length of the curves in Figure 2a is due to running the landscape model to completion (as described above).

---

## Author Comment (AC3) · 29 Dec 2018

*This manuscript addresses the "peak water" concept associated with glacier response to climatic warming. As reviewed in the introduction to the manuscript, this concept was described in two review articles and has been studied empirically in a number of site-specific studies. Although the empirical studies generally confirmed the conceptual model in broad terms, two fundamental questions arise from this body of literature: (1) what is the time scale over which the "peak water" cycle progresses, and (2) does the trajectory ultimately lead to reduced runoff.*

*To address these questions, the authors combined a numerical model of glacier dynamics with a parameterized model of vegetation succession and its influence on runoff. They applied the model to glaciers within simplified valley geometries for scenarios representing various combinations of bed slope, vegetation type, and rates of vegetation development for two different climate types and two different climate change scenarios.*

*The simulations confirmed that basin runoff ultimately decreases relative to pre- warming conditions. For scenarios without vegetation development, this decrease results from the surface lowering associated with glacier thinning and retreat, and the subsequent reduction in precipitation. Development of vegetation results in greater reductions in basin runoff. The magnitude of and time to "peak water" were greatest for continental glaciers with shallow bed slopes and lowest for steep maritime glaciers.*

*Overall, this is an interesting and relevant study. However, the conclusions, at least in qualitative terms, could have been deduced fairly directly from the underlying assumptions and basic knowledge of glacier dynamics. I believe that some further analysis and more detailed consideration of vegetation dynamics and ecohydrology would strengthen the contribution of this work. Some specific comments follow.*

Thank you for your careful review of our manuscript and your constructive feedback. It is clear from your review, as well as the other reviews, that we need to better articulate the objectives and scope of our study (particularly in the title and introduction). As you point out below, we have not accounted for several processes that likely affect basin runoff over decadal time scales. We agree that these processes are important and that they should be discussed in the manuscript. However, our goal was to focus on what we feel are the key controls on basin runoff: basin topography, climate, and revegetation. Work on modeling glacier retreat has indicated that basin topography and climate are key factors determining retreat rates, and the effects of revegetation on basin runoff have not been systematically explored. Instead of incorporating all of the additional processes that affect runoff (which could potentially be papers on their own), we have included brief discussion of these processes and their potential impacts on runoff. In addition, we have more clearly justified our selection of parameters.

*1. There are additional processes by which annual runoff would decline in a warming climate that are not accounted for in the model. First, as pointed out by another reviewer,*

*recent literature suggests that a shift from snow to rain results in decreased runoff even with no change in the amount of precipitation. Second, increasing air temperatures would be expected to increase evapotranspiration, subject to soil moisture availability. A third reason that one would expect glacier retreat ultimately to reduce basin runoff is that evaporation/condensation from snow or ice is typically low and often dominated by condensation, whereas an unglaciated surface would lose water by evaporation.*

For the first point, please see our response to the first reviewer. To the second point, we assume that the impacts to ET from changes in vegetation communities and biomass far outweigh changes driven by climate warming. This assumption is supported by Barnett et al., 2005, who found that increases in ET associated with climate warming are attenuated in snowmelt-dominated regions of the globe. The third point raised by the reviewer is important, however we feel that we have accounted for this effect as the changing runoff ratios account for net changes in evapotranspiration from moving from glaciated to vegetated terrain and thus include changes in condensation.

Barnett, T. P., Adam, J. C., and Lettenmaier, D. P.: Potential impacts of a warming climate on water availability in snow-dominated regions, Nature, 438, 303 EP –, https://doi.org/10.1038/nature04141, 2005.

*2. The scenarios represent glacier retreat followed by vegetation succession. However, retreat can also result in formation of lakes, which can accelerate glacier retreat and would ultimately provide an additional mechanism for reduced basin runoff via evaporation (Moyer et al., 2016). While it is likely not feasible to incorporate lakes into the model, this point should be acknowledged.*

This is a good point, and indeed lake-calving glaciers are often some of the fastest retreating glaciers (Larsen et al., 2007). Lakes could be incorporated into the model by using basin topography that has overdeepenings and then forcing faster retreat through the overdeepenings, although the processes driving calving retreat are poorly understood (e.g., Benn et al., 2007). Incorporating lakes in a systematic way is challenging, though, because the glacier evolution will depend on the location, depth, and length of the lake(s). Moreover, although evaporation from lakes will tend to reduce basin runoff, the formation of a lake prevents the development of a forest and will tend to increase basin runoff. In other words, evapotranspiration from a forest is being replaced with evaporation from a lake. With that in mind, we briefly mention the potential impact of lakes in the revised paper.

Benn, D.I., C.R. Warren, and R.H. Mottram (2007), Calving processes and the dynamics of calving glaciers, *Earth Sci. Rev.*, 82, 143-179.

Larsen, C.F., R.J. Motyka, A.A. Arendt, K.A. Echelmeyer, and P.E. Geissler (2007), Glacier changes in southeast Alaska and northwest British Columbia and contribution to sea level rise, *J. Geophys. Res.*, 112(F01007), doi:10.1029/2006JF000586.

*3. The model does not accommodate the development of a supraglacial debris layer, which can reduce meltwater generation and the rate of glacier retreat. See Frans et al. (2016). This point should at least be addressed as a discussion point if not incorporated into the model.*

We agree that a supraglacial debris layer can reduce meltwater generation and the rate of retreat, the latter of which has also been nicely demonstrated by Anderson and Anderson (2016) and Kienholz et al. (2017). With respect to our study, the challenge with including debris cover is that it depends strongly on bedrock lithology and therefore adds yet another parameter (erodibility) and would distract from the key questions that we are addressing. While not including it in the model, we have mentioned the potential impacts of debris cover on glacier recession in the revised manuscript.

Anderson, L.S. and R.S. Anderson (2016), Modeling debris-covered glaciers: response to steady debris deposition, *Cryosphere*, 10, 1105–1124, doi:10.5194/tc-10-1105-2016.

Kienholz, C., R. Hock, M. Truffer, P. Bieniek, and R. Lader (2017), Mass balance evolution of Black Rapids Glacier, Alaska, 1980–2100, and its implications for surge recurrence, *Front. Earth Sci.*, 5(56), doi:10.3389/feart.2017.00056.

*4. The analysis focuses on annual runoff, and the authors appropriately acknowledge the importance of considering seasonal runoff variations, particularly in late summer. This discussion could be extended by commenting on the relative magnitude of glacier contributions to seasonal and annual runoff (e.g., as a fraction of total runoff). Good references to draw upon are Frans et al. (2016) and Naz et al. (2014), both of which analyzed effects of glacier retreat on seasonal runoff.*

Thank you for this suggestion; we acknowledge this issue in the revised manuscript. Figure 6 illustrates how the proportion of (non)glacier runoff varies over decadal time scales with no vegetation. The proportion of glacier runoff on seasonal timescales should follow similar trends because, although the summer runoff from a glacier will increase during glacier retreat, the proportion of the basin that is occupied by glacier ice also decreases. Perhaps more important is the interannual fluctuations that occur and whether large interannual fluctuations occur when the runoff is near "peak water". We now mention in the introduction that we are modeling the "base flow" upon which seasonal and interannual variations are superposed. See also response to next comment.

*5. The climate scenarios do not include decadal fluctuations, which can complicate peak water cycles – e.g., by generating transient periods of glacier advance, at least early in the warming phase. See, for example, Figure 4 in Clarke et al. (2015) and Figures 8 and 9 in Frans et al. (2016). Also, the magnitude of glacier runoff varies interannually, being greater in warm/dry years than in cool/wet years. See, for example, Naz et al. (2014). This compensating effect is an important aspect of glacier contributions to basin runoff that is not captured in the model.*

We agree, and in some cases interannual variability in runoff may be more significant than long term trends (e.g. O'Neel et al., 2014). The net effect of interannual and decadal variability is an interesting question. Glacier retreat is primarily controlled by long time-scale fluctuations in climate, but short time-scale fluctuations could produce complex, nonlinear relationships between climate and runoff. For example, a series of cool/wet years may slow down a glacier's rate of retreat, causing it to be farther from equilibrium with the long time-scale climate trends (e.g., see Christian et al., 2018) and perhaps more susceptible to anomalously high melt rates in subsequent years. We also now mention this issue in the revised manuscript.

Christian, J.E., M. Koutnik, G. Roe (2018), Committed retreat: controls on glacier disequilibrium in a warming climate, *J. Glaciol.*, 64(146), 675-688, doi:10.1017/jog.2018.57.

O'Neel, S., E. Hood, A. Arendt, and L. Sass (2014), Assessing streamflow sensitivity to variations in glacier mass balance, *Clim. Change*, 123(2), 329-341, doi:10.1007/s10584-013-1042-7.

*6. The model scenarios are rather abstract, and I would encourage the authors to make a more structured effort to "map" the model scenarios into the real world. The authors should consider how they might synthesize their model results with results from the literature to develop a more nuanced conceptual model than those proposed by Jansson et al. (2003) and Moore et al. (2009).*

Thank you for this suggestion. Our model scenarios are indeed abstract, as our goal is to determine what controls the variations in basin runoff and the sensitivity of these variations to bedrock topography, climate, and vegetation rates. The metrics that we focus on (i.e., peak runoff, time to peak runoff, time to preretreat runoff, and end runoff) describe the shape of the hydrographs, and the sensitivity of these metrics to model parameters are what make our model more nuanced than those proposed by Jansson et al. (2003) and Moore et al. (2009). We have clarified this point in the revised manuscript.

*7. Related to the preceding comment, the analysis does not consider the covariation of vegetation succession, climatic regime and elevation, or their influences on runoff generation. The authors cite only two papers to support the range of runoff ratios and three papers to support the parameterized model of landscape evolution. The authors should review a broader selection of papers to provide a better framing of their vegetation scenarios. A selection from the last five years includes Wietrzyk et al. (2018), Fickert et al. (2017), Whelan and Bach (2017), Eichel et al. (2015), Klaar et al. (2015), Cowie et al. (2014) and Mizuno and Fujita (2014).*

We agree and thank the reviewer for the thorough and relevant literature provided. We have included a number of new references that help to frame the hydrological model in the revised manuscript (see also response to reviewer #1).